# Minimal vertical transport of microplastics in soil over two years with little impact of plastics on soil macropore networks
Roman B. Schefer [1], John Koestel[2] & Denise M. Mitrano [1] ✉

Plastics used in agriculture improve productivity and resource efficiency. As they fragment over time, microplastics are unintentionally released into soil, raising concerns regarding long-term implications for soil structure and fertility. Here we investigated microplastics transport and their impact on soil structure through a two-year field experiment. 45 re-packed soil columns were installed with three treatments: indium-doped polyethylene terephthalate fragments or fibers in the top 2 cm and a control with no microplastics. Soil pore structure was monitored with X-ray tomography, and microplastics vertical transport was assessed via the indium tracer. With time macropore volume, biopore fraction and critical pore diameter increased independent of microplastic addition. Microplastic transport was minimal, with only ~1% reaching below 8 cm soil depth in two years. This experimental design, simulating natural soil conditions, suggests that microplastics have a negligible influence on soil macropore architecture and its transport rate is limited in the short term.

Amidst rapid industrialization and a growing global population, responsible resource management is increasingly emphasized, particularly in critical areas such as food and water. In agriculture, ensuring robust crop yields and efficient soil and water management are paramount. Plastics have emerged as a significant contributor, seamlessly integrating into modern agricultural practices to support these goals. However, it is important to recognize that the immediate advantages of plastic utilization may come at the expense of long-term sustainability goals[1]. Plastics enter these systems through practices including the use of sewage sludge as fertilizer[2] and their application as mulch films, translucent sheeting for greenhouses, and soil conditioners[3,4]. Atmospheric deposition onto agricultural land further contributes to plastic contamination burdens[5]. Plastics in soils can therefore exist across a broad size spectrum and diverse polymer compositions. Microplastics (MPs, <5 mm), being small in size, pose a challenge for removal once they enter the ecosystem. While studies have begun to examine MPs concentrations in soils, reported levels vary widely[6–9].

One important aspect of soil fertility is soil structure, that is, the abundance, arrangement and connectivity of pores[10]. Pores of different diameters are associated to specific soil properties. For example, macropore networks (pore diameters >30 μm) are fundamental for gas exchange between soil and atmosphere[11], discharge of water under high intensity rainfall events[12] and soil susceptibility to preferential flow[13]. They are crucial for soil functions such as providing a growing medium for plants[14] and habitats for soil fauna[15]. Macropore networks are not stable over time[16,17].

Recent research suggests that their evolution is predominantly driven by soil fauna and plant roots[18].

The persistence of (micro)plastics in the environment is primarily determined by their molecular backbone and its resistance to enzymatic cleavage, which governs their biodegradability. Therefore, MPs made of conventional non-biodegradable materials are likely to be persistent in the environment and may trigger unforeseen ecological changes[19]. Our understanding of the multifaceted consequences of MPs on soil quality, plant growth, and productivity is still in its early stages, and initial studies have demonstrated indirect effects of MPs in soil systems[20–23]. However, these investigations have often been limited in duration, lasting less than two months, and were primarily conducted in laboratory settings, focusing on aspects such as soil aggregation[20–23]. Long-term field studies (duration of 2 years or more) have focused mainly on the accumulation of MPs in agricultural soils, with only few examining their impact on soil properties or realistic transport rates. MPs were found to alter soil pH, moisture retention, and nutrient availability, notably increasing nitrogen levels while decreasing the C/N ratio. Furthermore, MPs disrupted the structure of soil microbial communities, promoting the growth of pathogenic microbes and reducing the diversity of beneficial microorganisms. These alterations in soil physical and biological properties were linked to reduced crop productivity and water use efficiency, with plastic residues also inhibiting root growth and water movement[6,24–28]. However, the effects of MPs on soil morphology over extended periods in natural field settings remains less explored. To better

[1]Environmental Systems Science Department, ETH Zürich, Universitätstrasse 16, 8092 Zürich, Switzerland. [2]Soil Quality and Soil Use, Agroscope, Reckenholzstrasse 191, 8046 Zürich, Switzerland. ✉e-mail: denise.mitrano@usys.ethz.ch

assess the impact of MPs on soil morphology, quantitative analyses over more extended exposure periods are essential, necessitating in situ studies under field conditions. Investigations into macropore networks may provide insights into the possible impacts of MPs on burrowing behavior of soil fauna or plant growth, as discussed by Khalid et al. [29]. Furthermore, quantifying MPs transport through soil depth over time will help improve risk assessments by providing insight on the time-scales MPs move through the soil profile and under which conditions.

Previous research suggested a strong association between changes in macroporosity and alterations in soil aggregation induced by MPs. For instance, an increase in pores larger than 30 μm was observed in soils treated with polyester fibers (0.3 wt.%) during both field and pot experiments, indicating a potential enhancement of soil aggregation[30]. Varying effects of polyacrylonitrile fibers were reported in pot incubation experiments under natural field conditions, with lower concentrations (0.001 & 0.01 wt.%) decreasing macroporosity and higher concentrations (0.1 wt.%) increasing it, while the addition of polyethylene fragments showed no significant effects[31]. Conversely, polyester fibers (0.5 wt.%) increased macroporosity, likely through improved soil aggregation, as observed in pot experiments[32]. Despite these valuable insights, a consensus on the overall impacts of MPs on soil macroporosity remains elusive, with most studies concentrating on short-term effects and using indirect methods to assess macroporosity through soil hydraulic properties (e.g., tension tables or pressure plate apparatus)[10,23,30–34]. Direct image analysis for quantifying macroporosity and connectivity measurements have been relatively underexplored in this context. Advanced techniques such as X-ray computed tomography (X-ray CT) offer a promising avenue to quantify changes in soil macropore structure over time, as well as assess macropore connectivity, biopores and particulate organic matter (POM), providing a comprehensive understanding of MPs' potential impact. This capacity to monitor soil structural evolution is crucial for understanding how changes in soil pore structure may be modified, along with associated soil functions that are important for crop production and organic carbon and nutrient cycling[17,35,36].

The transport of MPs through soil depends on several factors, including particle size, morphology, polymer chemistry and soil texture[37–39]. Literature on colloid transport in porous media has shown that particles in the size range of 1 μm tend to be more mobile than smaller or larger ones. Larger particles (>1 μm) are prone to straining and physicochemical interactions (van der Waals forces, electrostatic interactions, and hydrophobic effects) that hinder their movement, while smaller particles, such as nanoparticles, may encounter increased retention due to electrostatic and surface interactions with the porous matrix[40–42]. Nanoplastic (<1 μm) may, however, also adsorb to soil colloids that are more mobile and may be transported in this fashion. Additionally, for MPs, particles with complex, nonspherical shapes exhibited higher adhesion to porous media, reducing their mobility compared to more spherical or simple shapes[43,44]. Moreover, factors such as ploughing, soil morphology (including cracks and preferential water flow), bioturbation, and rainfall intensity considerably contributed to the transport of microplastics[37,45]. Bioturbation by earthworms, in particular, has been shown to enhance the transport of MPs down to deeper soil layers as a consequence of burrow formation[46,47]. Despite the importance of understanding vertical MPs transport, conducting research in this area presents significant challenges. Sampling and extracting MPs from complex soil matrices, along with the necessity of specialized analytical techniques, have made such studies inherently difficult[39,48–50]. However, promising methods such as doping MPs with trace metals have emerged to aid in tracking their mobility more precisely and easily[51]. This approach allows for the use of established methods (e.g., Inductively Coupled Plasma Mass Spectrometry (ICP-MS)) to measure metals as a proxy for the plastics in complex matrices like soil[43,52–54].

In this current study, we aimed to explore the effects of MPs on soil pore structure by investigating the transport of MPs fragments and fibers in soil under field conditions and assessed its impact on the formation of soil macropore network structures over two years (Fig. 1). This involved spiking indium-doped MPs (0.2 wt.%) at the top of soil columns that had been

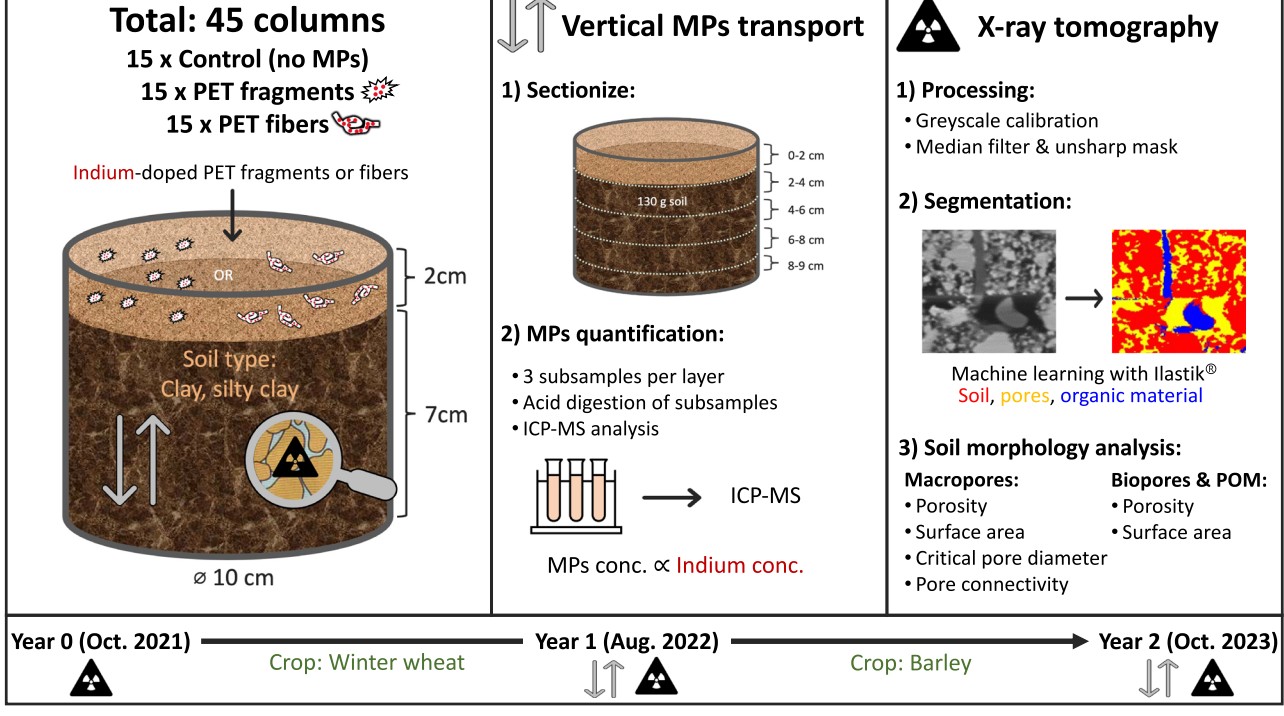

**Fig. 1 | Schematic of the overall experimental design.** A total of 45 soil columns were installed in an agricultural field: 15 control columns without MPs, 15 columns with indium-doped PET fragments, and 15 columns with indium-doped PET fibers. The vertical transport of MPs over two years was investigated by (1) sectioning the column to five layers, and (2) MPs quantification which involved acid digestion of subsampled layers and ICP-MS analysis of the indium concentrations, which were proportional to the MPs concentration. The evolution of soil pore structure over two years was analyzed via X-ray tomography. This involved (1) raw X-ray image processing, (2) segmentation into soil, pores, and organic material, and (3) subsequent analysis of soil morphology measures. Sampling timepoints for X-ray tomography were at the start (year 0), year 1, and year 2, while vertical transport analysis was conducted only after year 1 and year 2. Winter wheat was cultivated between year 0 and year 1, and barley was cultivated between year 1 and year 2.

packed with soil. The packed soil columns were then placed in an arable agricultural field following typical cereal crop cycles (wheat and barley). They were exposed to natural weather conditions and had open bases connecting them to the subsoil. Additionally, the columns were perforated along the sides to allow for exchanges with the surrounding soil environment. We utilized X-ray CT to monitor the evolution of soil macropore structure from year 0 to year 1 and year 2, and employed ICP-MS to follow the vertical transport of MPs, both at the one- and two-year time points. Collectively, our study addressed two primary objectives: i) evaluate the impact of MPs on soil morphology, specifically assessing any observable effects on the macropore network structure, including the abundance of biopores, and ii) uncovering the rate at which MPs are transported through the soil matrix, a crucial step in assessing their environmental implications. As a corollary aim, we analyzed the overall change in soil morphology over two years. Our study focused on PET fragments and fibers, common forms of MPs as model materials to explore MPs pollution in soils.

## Results

For soil morphology analysis of the columns, three different regions of interest (ROIs) within the soil column were examined. ROI 1 encompassed nearly the entire soil column, measuring 6.5 cm in height, and was further divided into two subregions: ROI 2, which represented a small part (1 cm

height) of the top of the core where MPs were initially added at time zero, and ROI 3, which consisted of the lower 4 cm of the core where we anticipated less MPs accumulation. Separately, vertical transport of MPs in the soil columns was assessed across five layers, which were physically sectioned and further analyzed for MPs content. The uppermost four layers were each 2 cm thick, and the lowest layer varied between 0 and 1 cm, depending on the compaction of the soil over time (Fig. 1). Further details regarding the ROI for X-ray CT and the sectioning of the soil columns can be found in the methods section. Exact p-values, point estimates, standard errors and degrees of freedom of statistical analysis of all morphology measures are summarized in Supplementary Section S11 (SI).

## Temporal evolution in soil morphologies of controls

Using our dataset from the control columns (no MPs), we investigated the temporal evolution of soil morphologies in ROI 3, where we had an initial soil structure resembling a freshly prepared seedbed by avoiding the less natural sieved soil layer on top (ROI 2). Notable changes in soil structure in ROI 3 were observed over a period of two years. While macroporosity remained similar between year 0 and year 1 ($-0.73\%$, $p = 0.67$), it decreased by approximately 8% in year 2 ($p < 0.001$) (Fig. 2a). Additionally, the surface area of macropores showed an overall decrease over time, with a reduction of around 50% ($p < 0.001$) (Supplementary Fig. S1a, SI). In year 0, 91.4% of

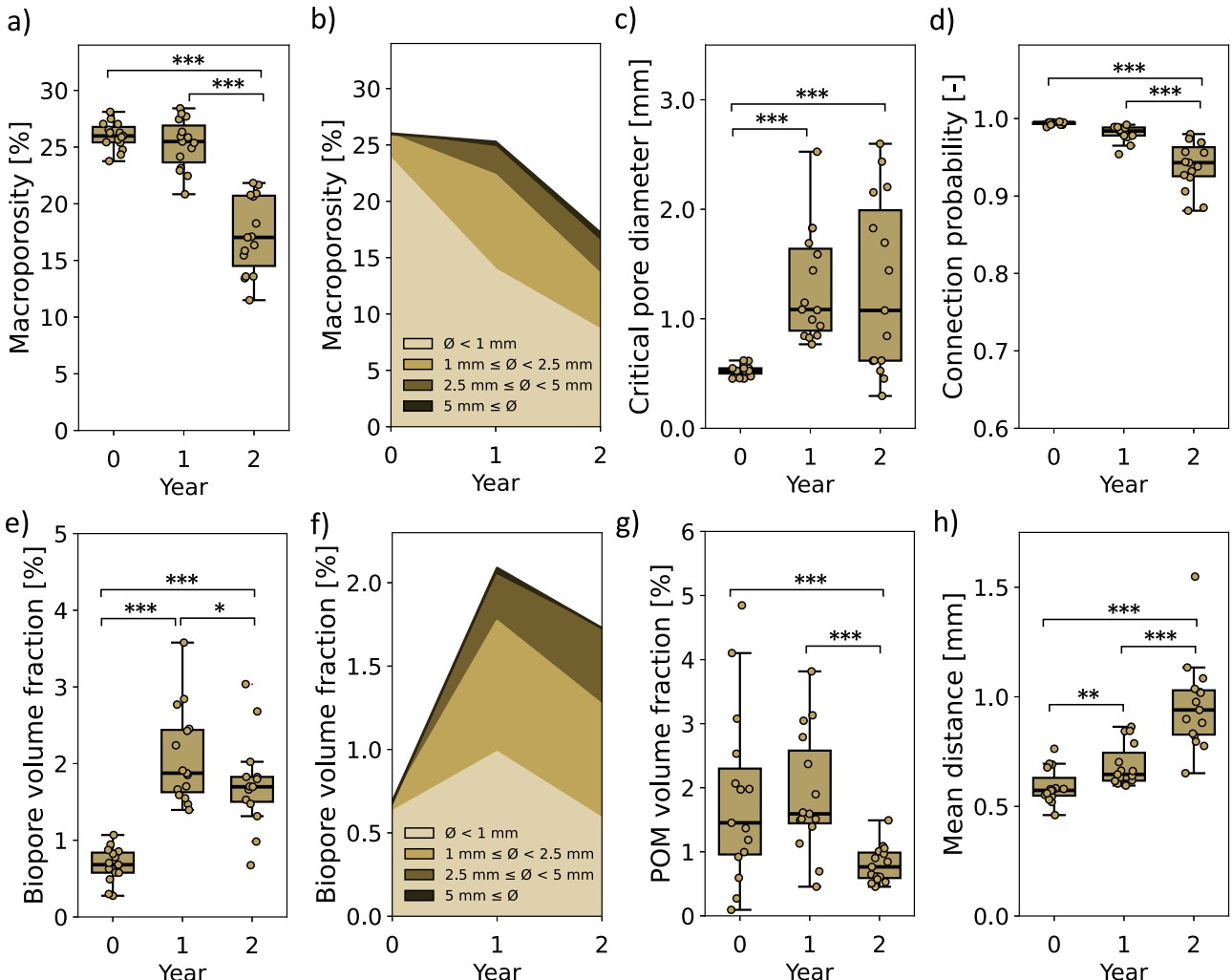

**Fig. 2 | Summary of the temporal soil morphology evolution in ROI 3 of the control samples ($n = 15$).** Including macroporosity (**a**), pore size distribution of the macropores with colors indicating size ranges (**b**), critical pore diameter of macropores (**c**), connection probability (**d**), the evolution of biopore volume fraction (**e**), pore size distribution of biopores (**f**), POM volume fraction (**g**), and mean distance of soil matrix to next aerated macropore connected to the top (**h**). Boxplots show dots: individual data points, line: median, box: lower and upper quartile, whisker: highest and lowest value, dots outside whisker: outlier. Statistical significance is indicated as follows: $p$-value < 0.001 ***, <0.01 **, and <0.05 *.

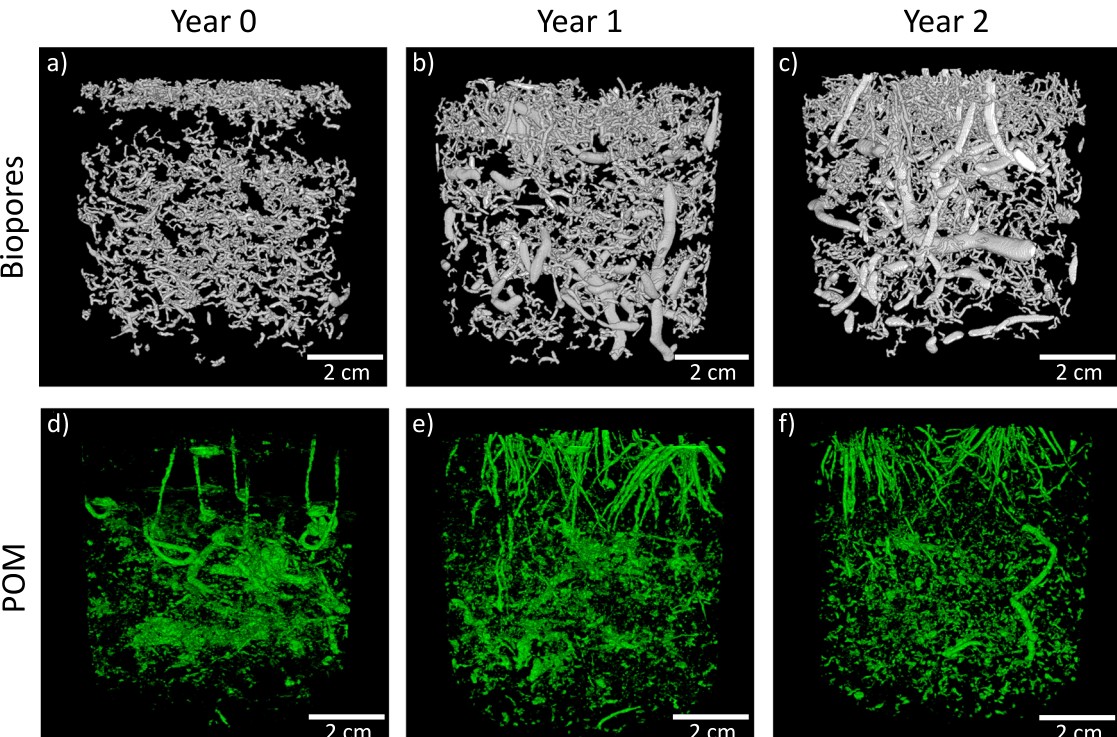

**Fig. 3 | Images of resolvable biopores and POM of ROI 1 of a control column without MPs addition.** Biopores in top row (white) and POM in bottom row (green). Panels (**a–c**) correspond to biopores at year 0, year 1, and year 2, respectively. Panels (**d–f**) correspond to POM at year 0, year 1, and year 2, respectively.

macropores were smaller than 1 mm, 8.2% were between 1 mm and 2.5 mm, and 0.3 % were between 2.5 mm and 5 mm (Fig. 2b). By year 1, larger macropores (1 mm to 5 mm) had formed, and the proportion of smaller pores (<1 mm) had decreased. Specifically, 55.2% of pores were <1 mm, 33.1% were between 1 mm and 2.5 mm, 9.7% were between 2.5 mm and 5 mm, and 2.0% were ≥5 mm. In year 2, macroporosity further decreased, with a similar size distribution to year 1, with more macropores larger than 2.5 mm. The connectivity measures also changed over time. The critical pore diameter increased by 2.5 times, from 0.52 mm in year 0 to 1.30 mm in year 1 ($p < 0.001$) and remained unchanged in year 2 (Fig. 2c). Conversely, the connection probability of the macropores decreased over time, reaching 0.94 by year 2 ($-0.055$, $p < 0.001$), decreasing from an initial value of 0.99 (Fig. 2d). Further, the mean distances of the soil matrix to the next aerated macropore with connection to the soil surface increased ($+0.34$ mm, $p < 0.001$) (Fig. 2h).

Examining biopores separately revealed changes in their volume fraction over time. The biopore volume fraction increased from 0.7% in year 0 to 2.1% in year 1 ($+1.41\%$, $p < 0.001$) and remained largely unchanged in year 2 ($-0.39\%$, $p = 0.023$), with a slight reduction (Fig. 2e). This trend was likewise evident in the three-dimensional representation of temporal biopore evolution in the whole ROI 1 of a control column (Fig. 3a–c). The biopore surface area in ROI 3 followed a similar pattern, with an additional slight reduction from year 1 to year 2 ($-0.006$ m², $p < 0.001$) (Supplementary Fig. S1b, SI). The pore size distribution of biopores increased towards larger biopores over time (Fig. 2f). In Year 0, 91.8% of biopores were smaller than 1 mm, and 8.2% were between 1 mm and 2.5 mm. In year 1, larger biopores (1 mm to 5 mm) became more prevalent, while the proportion of smaller biopores (<1 mm) decreased. Specifically, 47.1% of biopores were <1 mm, 37.6% were between 1 mm and 2.5 mm, 13.1% were between 2.5 mm and 5 mm, and 2.2% were ≥5 mm. By year 2, the overall bioporosity had decreased. However, the size distribution of biopores remained similar as observed in year 1, with an increase in the size fraction ranging from 2.5 mm to 5 mm and a decrease in the fraction smaller than 1 mm.

The analysis of particulate organic matter (POM) from X-ray images showed that POM levels increased slightly between years 0 and 1 ($+0.25\%$, $p = 0.39$) with little evidence, but decreased by year 2 ($-0.84\%$, $p < 0.001$) (Fig. 2g). The surface area of POM followed a similar trend (Supplementary Fig. S1c, SI).

### Minimal impact of MPs on soil morphologies

Collectively, minimal differences in soil morphologies were observed between control columns and those amended with PET MPs fragments or fibers over the two-year column incubation. The primary observations for initial differences came from ROI 2, which had the highest concentration of MPs, revealing significant changes in the porosity of the macropores and biopores. Specifically, macroporosity was marginally higher in soils containing PET MPs fragments ($+2.8\%$, $p = 0.003$) and fibers ($+4.8\%$, $p < 0.001$) at year 0 (see Fig. 4a, b). However, by years 1 and 2, these differences were less pronounced between the treatments with very uncertain evidence. The biopore volume fraction was slightly higher in fragment-treated soils compared to the control in all years (year 0: $+0.39\%$, $p = 0.004$, year 1: $+0.35\%$, $p = 0.012$, year 2: $+0.43\%$, $p = 0.007$), while fibers only showed a moderately higher biopore volume fraction than the control in year 0 ($+0.34\%$, $p = 0.019$) (Fig. 4c). The same trend was observed for the biopore surface area although with very uncertain evidence (Fig. 4d). Despite some minor differences, there was little to no evidence for a clear effect of MPs shape (fragments vs. fibers) on soil morphologies, with the observed effects being minimal. These general slight and initial differences in soil morphologies appear to have diminished over time or were overshadowed by other soil processes such as bioturbation, root growth, wet/dry cycles, and freeze/thaw cycles.

In ROI 3, little to no differences in soil morphologies were observed between the control and MPs-treated soils. After 2 years, the fragment treatment showed a slight increase in biopore volume fraction ($+0.44\%$, $p = 0.012$) and biopore surface area ($+0.0038$ m², $p = 0.011$) compared to the control (see Supplementary Fig. S6, SI). Additionally, slight changes due to the inclusion of fragments were evident in macropore surface area at year

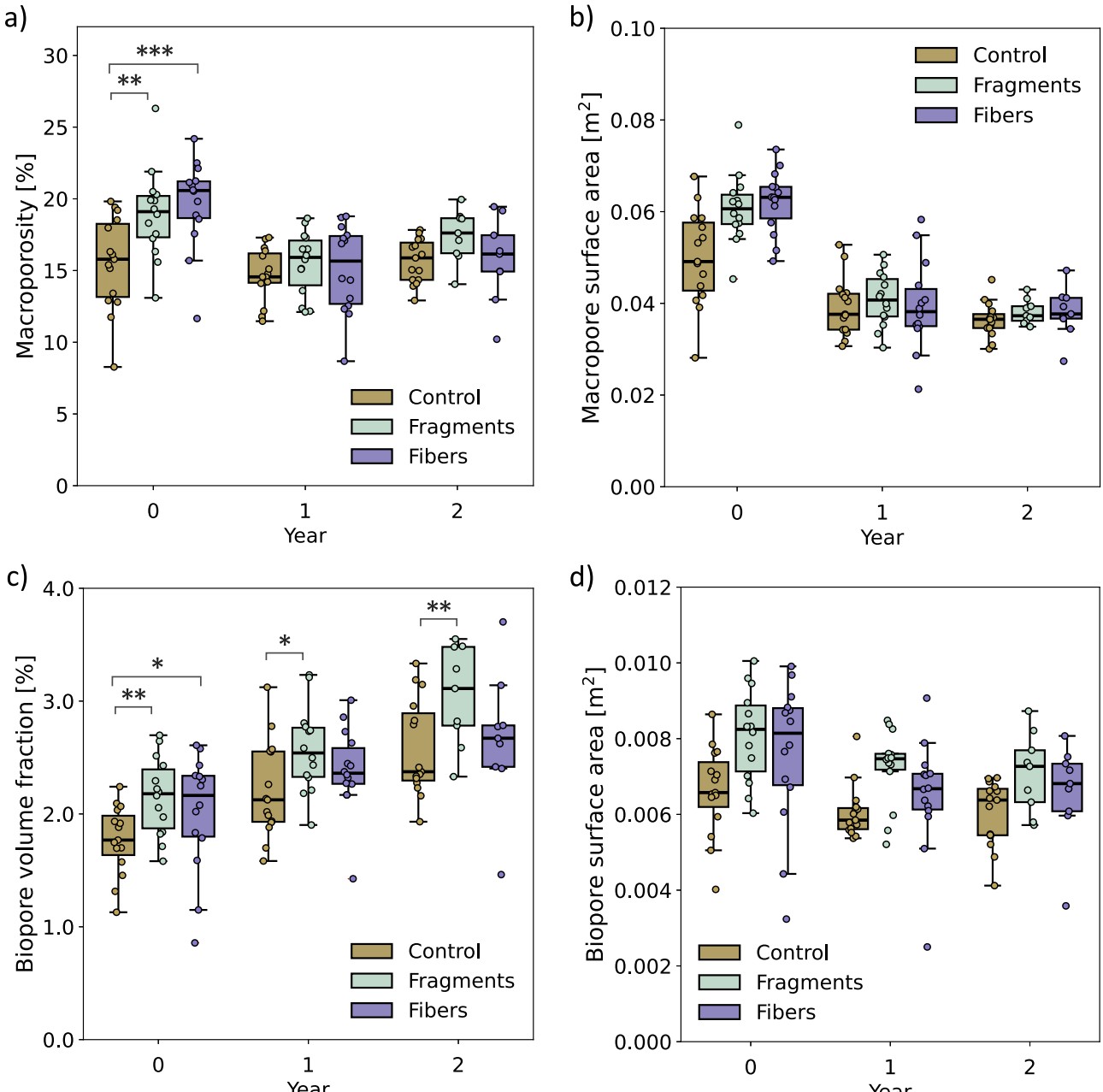

**Fig. 4 | The evolution of soil morphologies in ROI 2.** Macroporosity (**a**), specific surface area of macropores (**b**), biopore volume fraction (**c**), and biopores surface area (**d**). Each treatment is represented by a distinct color: control (beige), MPs fragments (cyan), and MPs fibers (violet). For the box plots; dots: individual data points, line: median, box: lower and upper quartile, whisker: highest and lowest value, dots outside whisker: outlier. Statistical significance is indicated as follows: *p*-value < 0.001 ***, <0.01 **, and <0.05 *.

0 ($-0.017$ m², $p = 0.029$) and year 2 ($+0.026$ m², $p = 0.003$), as well as in critical pore diameter at year 2 ($-0.53$ m², $p = 0.047$). Despite the low evidence suggesting that MPs influence soil morphologies in ROI 3, the effect sizes were negligible. This suggests that the concentration of MPs present in ROI 3 did not noticeably affect the overall soil morphology, although it remains possible that MPs influenced pore network structures at smaller scales, which were not resolved in our X-ray images. All soil morphology measures for the different treatments and ROIs over the two years are shown in the SI, Supplementary Sections S2–S4.

**Vertical transport of MPs fragments and fibers**

MPs concentrations were notably greatest in the two uppermost layers of the soil profile, gradually decreasing with deeper depths showing a significant layer effect ($F = 640.87$, $p < 0.001$) (Fig. 5). Both MPs fragments and fibers exhibited similar trends across the two-year monitoring

period, with slightly increased MPs concentration observed below depths of 4 cm after two years (year effect: $F = 20.31$, $p < 0.001$). As expected, the highest concentration was found in the uppermost layer, which was initially spiked with MPs, but MPs were also observed in high concentrations in the second layer as well. 1% of MPs reached the lowest layer following one- and two-year incubation periods. There appears to be a difference in transport of fragments versus fibers (replicate effect: $F = 9.72$, $p = 0.0052$), especially in layers 2 and 3 (see significance annotations in Fig. 5), but the direction of the effect varied. Notably, measured MPs concentrations in all sampled soil layers exceeded the method detection limit (MDL) of 1.52 mg per 100 g soil. Moreover, a high recovery rate of initially added MPs was achieved after both one and two years. Specifically, fragment recovery rates were $96.6 \pm 1.5\%$ and $83.8 \pm 2.9\%$ after one and two years, respectively. Fiber recovery rates were $97.2 \pm 1.7\%$ after one year and $86.1 \pm 5.6\%$ after two years.

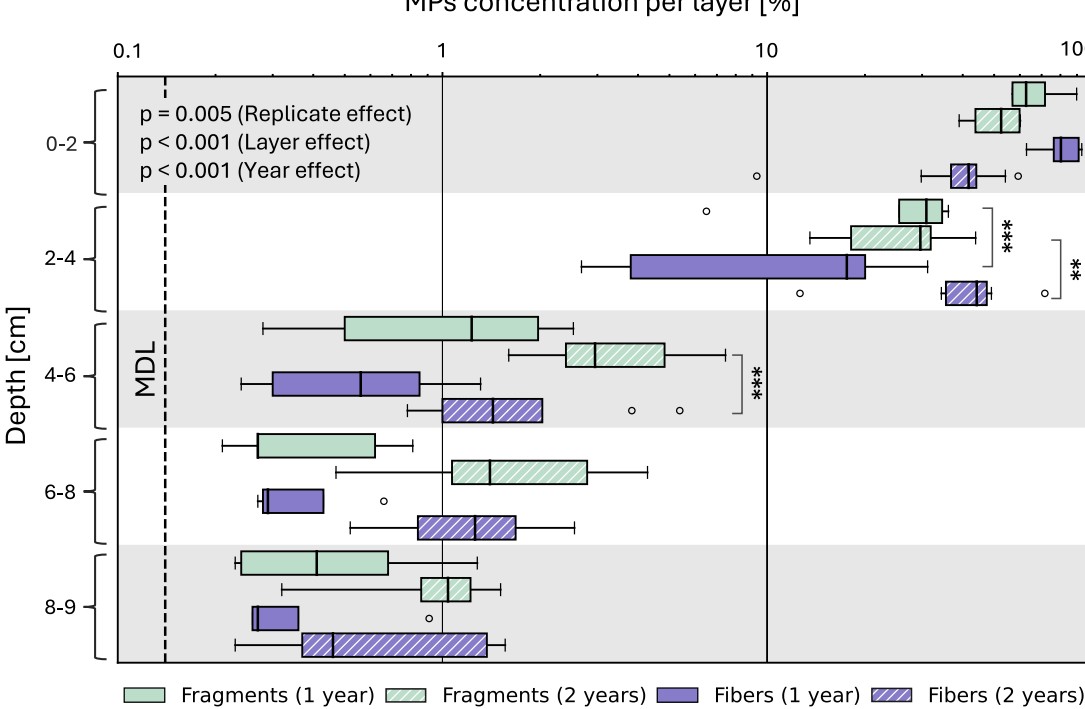

**Fig. 5 | Vertical transport of MPs.** MPs fragments (cyan) and fibers (violet) after one- (solid bars) and two- (hashed bars) year incubation in the field. The x-axis is presented on a log scale, which compresses higher values and can affect the proportional appearance of the boxplots. Method detection limit (MDL) is shown as black dashed vertical line. In the box plots, line: median, box: lower and upper quartile, whisker: highest and lowest value, empty dots: outlier. Statistical significance for effect of replicate is indicated as follows: $p$-value < 0.001 ***, <0.01 **, and <0.05 *.

## Discussion

### Temporal evolution in soil morphologies of controls

The evolution of soil structure over time in the control samples, particularly in ROI 3, emphasizes the dynamic nature of soil systems. Significant changes were observed in soil morphologies with a notable decrease in macroporosity and macropore surface area. The temporal evolution of pore size distribution showed an initial formation of larger macropores by year 1, followed by a reduction in overall macroporosity by year 2. This pattern mirrors typical soil consolidation and stabilization processes post-tillage and seedbed preparation, where initial soil disturbance leads to a temporary increase in porosity, followed by gradual compaction and stabilization[55]. Consistent with this trend, the estimated bulk density of the entire soil column increased over time, from $0.86 \pm 0.05$ g cm$^{-3}$ after year 1 to $0.98 \pm 0.07$ g cm$^{-3}$ after year 2. Despite the overall decreasing macroporosity, the observed trends towards larger macropores and increased biopore volume fractions and biopore surface areas highlight the ongoing reorganization of soil structures in response to environmental factors. This aligns with previous studies where bioturbation by soil macro-fauna, such as earthworms, was shown to be a dominant factor in soil morphology evolution[17,18,56]. These larger macropores and biopores, coupled with high connection probabilities, facilitate greater gas exchange and improve soil aeration. The increase in critical pore diameter is associated with higher saturated hydraulic conductivity[57] and enhanced water infiltration, thereby influencing the soils' ability to maintain its structure and function over time. However, the increased mean distances from the soil matrix to the nearest aerated macropore suggest a slight worsening of the soil matrix aeration conditions.

The decline in POM over the study period reflects shifts in organic matter inputs and decomposition rates. The high POM level at year 0 is likely due to decomposed roots and plants from initial soil packing and germinating winter wheat seeds (Fig. 3d). By years 1 and 2, new roots were visible (Fig. 3e, f), with more decomposed roots by year 2 due to a longer interval between barley harvest and the X-ray scan compared to the interval for winter wheat in the first year.

### Minimal impact of MPs on soil morphologies

Contrary to the initial concerns regarding significant alterations in soil structure due to the presence of MPs, our study reveals that changes are minimal and transient even at high MPs concentration of 1 wt.%, which may be relevant for hotspots of contamination or regions with cumulative plastic application over time. Initial differences in macroporosity in MPs-treated soils were observed in ROI 2, where the soil had been spiked with MPs. However, the differences did not persist over two years. This transient effect implies that while MPs might have an impact on soil structure following packing of sieved soil, natural processes occurring within the soil such as bioturbation, root growth, and seasonal cycles (wet/dry and freeze/thaw cycles) might mitigate these impacts over time. The lack of sustained differences in soil morphology between control and MPs-amended soils suggests that soil systems possess a certain resilience, which is crucial for maintaining soil health and function[58]. However, these findings do not conclusively rule out negative impacts of MPs on soil and its biological processes. Microporosity may still be affected, and the uptake, bioaccumulation, and trophic transfer of MPs could pose hazards to organisms. Additionally, leached chemicals and metals from plastics could harm soil biota[7,59–61]. Therefore, ongoing research and monitoring are essential to mitigate environmental risks from (micro)plastics. Future studies should focus on the mechanisms of MPs interactions with soil constituents, soil biota, and nutrient cycles. Investigating MPs interactions with specific soil biota, along with the cumulative effects of varying chemistries, sizes (including nanoplastics), and concentrations over time, is crucial. Additionally, enhancing experimental setups with detailed analyses of soil microbial activity will improve our understanding of local conditions for microorganisms.

### Vertical transport of MPs fragments and fibers

The vertical transport of MPs was conclusively shown by measuring indium, which correlated with the MPs. Both MPs fragments and fibers exhibited a similar trend across the two-year monitoring period. The comparison of fibers and fragments revealed that their transport

dynamics depended on layer and year. However, in most layers there were no clear differences between shapes suggesting other factors have a stronger influence. While we were not able to disentangle the influence of bioturbation (from plant root growth and organisms) from advective transport explicitly, other studies indicated that MPs sizes like in our study are barely transported with infiltrating water[47,62,63]. We therefore hypothesize bioturbation to be an important factor in driving MPs transport through soil depth[46,47]. Furthermore, our observations of the biopores indicate that the presence of MPs did not significantly impact bioturbation. The greatest concentrations of MPs were found in the uppermost soil layer, which aligns well with the fact that MPs were initially added only to the top of the columns when packed. Still a considerably high concentration of MPs was observed in the second layer. This may be attributed to short-distance transport of MPs, but also to spatially heterogeneous soil settling resulting in an uneven interface between the MPs-spiked soil layer and the layer below. The movement of MPs observed beyond 4 cm depth after two years underscores the persistent, albeit relatively slow penetration rate compared to other contaminants[53]. In another study examining the transport of larger MPs (710–850 µm) by earthworms over a 21 days plot experiment, it was found that more than 50% of the added MPs were transported to depths below 7 cm[47]. This study included the insertion of four earthworms per plot, which could not escape, likely explaining the higher transport rate due to increased bioturbation compared to our finding of roughly 2% MPs to the same depths. It should be noted that the burrowing behavior of different earthworm species and other soil-dwelling organisms can deviate from the one of L. Terrestris[64]. It follows that also rates and directions of material redistribution could also differ.

The high recovery rates of MPs indicate a robust methodological approach. Given the methods' spiked recovery rates for fragments (96.2 ± 5.5%) and fibers (95.9 ± 6.7%) as discussed in the method section, a certain loss of MPs from the columns was evident after the two-year incubation period (average loss after two years: 12.4% for fragments and 9.8% for fibers). While a portion of the MP loss can be attributed to the analytical process, such as soil material removed by the saw blade during soil core cutting and soil retained in column holes that was excluded from analysis, this alone does not fully explain the reduction in MPs recovery rates observed. It is likely that additional losses were due to MPs migrating vertically within the soil, being redistributed or laterally transported by soil fauna and roots, or being lost from the soil surface through wind and splash erosion[60,65–68]. These insights are critical for understanding the environmental persistence of MPs and their potential to affect soil ecosystems over extended periods.

## Conclusions

This study explored transport of MPs, their effects on soil morphology and the temporal evolution of soil morphology under field conditions. Quantitative analysis of MPs concentrations was ensured with the use of metal-doped plastics, which aided in providing robust results. Over the two-year period, a fraction of MPs were transported to deeper layers, likely due to bioturbation, but their shape did not significantly affect transport. This slow migration suggests that MPs could accumulate in the upper layers of soil, potentially creating localized pollution hotspots, while also offering an opportunity for targeted remediation efforts. We found significant changes in soil structure over time in the absence of MPs, characterized by a shift towards larger macropores, and an increase in biopore volume fraction and critical pore diameter. The addition of MPs did not alter these trends. While previous studies have raised concerns about potential impacts of MPs on soil structure, our findings revealed minimal effects on soil macropore and biopore network only in the initial phase of the incubation. This suggests that soil properties of seedbeds evolve towards a more settled and naturally structured soil with minor influence of the presence of MPs. Our study provides reassurance that MPs, at least at concentrations of 1% wt., may not disrupt soil structure formation and development as much as initially feared. However, higher MP concentrations in hotspots could still affect soil macropore networks and other critical soil properties, emphasizing the need

for continued monitoring and research on the long-term effects of MPs on various soil properties. Further, it is essential to acknowledge that our findings do not definitively rule out the possibility of MPs affecting soil quality parameters and important biological processes. Microporosity may still be altered by the presence of MPs, and there are potential risks associated with the uptake, bioaccumulation, and trophic transfer of MPs to organisms. Furthermore, chemicals and/or metals leaching from plastics may adversely impact soil biota. As MPs concentrations increase in the future, significant changes in the macropore network may emerge, and the further fragmentation of MPs to nanoplastics poses a risk of faster downward transport and additional negative implications in terms of irreversible pollution and biological hazard.

In light of these findings, reducing plastic use in agriculture and exploring sustainable alternatives are important goals. However, alternatives to plastics should undergo thorough testing and regulation before widespread adoption to ensure they do not introduce similar or new risks to soil ecosystems. By prioritizing sustainable solutions, we can mitigate the potential for irreversible soil pollution, reduce plastic-related ecological footprints and better safeguard agricultural ecosystems.

## Materials and methods
### Chemicals

Nitric acid (HNO$_3$, 65%) was purchased from VWR International Inc. Polyethylene terephthalate (PET #5997, density 1.4 g cm$^{-3}$ was purchased from Serge Ferrari Tersuisse AG (Luzern, Switzerland). Indium oxide nanoparticles (In$_2$O$_3$, 16–68 nm diameter) were purchased from Nanografi (Ankara, Turkey). ICP-MS standards of Rhodium (10 mg kg$^{-1}$) and Palladium (10 g L$^{-1}$) were purchased from Sigma Aldrich, and indium (1 g L$^{-1}$) from Alfa Aesar (Massachusetts, USA).

### Model metal-doped MPs fragments and fibers

Polyethylene terephthalate (PET) fragments (65–125 µm) and fibers (Ø 30 µm, 0.5–2,0 mm length) with an inclusion of ~0.2 wt.% indium were developed in house to assess the extraction workflow and vertical transport of MPs in soil, where the metal was measured as a proxy for plastic using ICP-MS[52]. The production of the indium doped PET fibers was performed according to the methods described in Frehland et al.[69], and the indium doped PET fragments were produced following the procedures detailed by Tophinke et al.[52]. Briefly, for pellet production, PET was melt-mixed with In$_2$O$_3$ nanoparticles in a 36 L/D twin screw extruder (Collin Lab & Pilot Solutions GmbH, Germany). Initially, 5 wt.% In$_2$O$_3$ was mixed with 95% PET, followed by compounding with 95% PET, resulting in an approximate 0.2 wt.% indium content. Extrusion conditions were 300 °C and 0.8 kg h$^{-1}$, preceded by drying the polymer for 8 h at 140 °C. The extruded polymer was cooled on a conveyor belt and pelletized. For MPs fragments, the pellets were ground using a rotor mill Pulverisette 14 (Fritsch, Germany) under cooling with liquid nitrogen. MPs fragments of 65–125 µm diameter were collected via sieving through stainless-steel sieves[52]. To produce MPs fibers, pellets were melted and fed through a single screw extruder to a melt pump, which delivered the polymer melt to a spin pack. The melt was then extruded into fibers, drawn at a ratio of 3.5 with a final winding speed of 700 m min$^{-1}$. To facilitate cutting, the fibers were annealed at 200 °C for 30 min to increase their crystallinity[69]. While polyethylene is the most common polymer in soils, mainly from agricultural mulching films, PET is still a good choice as a model for studying plastic pollution. PET fragments and fibers, often from textiles and packaging, are increasingly found in terrestrial environments[70,71].

### Agricultural field site

The agricultural field site was located in Zürich Reckenholz Switzerland (47°25'52.5"N 8°31'24.3"E) with dimensions measuring 11.5 m × 7 m. The soil at the site has been characterized by LUFA[72] and classified as deep gleyic Cambisol[73] with a loam texture (USDA texture classification) and sand, silt, and clay fractions of 0.305, 0.49 and 0.205 g g$^{-1}$, respectively. The organic carbon content was 0.015 kg kg$^{-1}$ and the soil had a

pH of ~4.97. A meteorological station from the Swiss Federal Office of Meteorology and Climatology (MeteoSwiss) is located within 200 m of the experimental site. Over the course of the two-year experimental period (from October 11, 2021, to October 12, 2023), the mean temperature was 10.9 °C, with a mean annual precipitation of 879.9 mm[74]. Detailed monthly temperature and precipitation data can be found in Supplementary Fig. S8.

### Field preparation, soil sampling and soil column preparation

Prior to the soil structure incubation experiment, the plot was used as a permanent grassland for more than 20 years. The site was prepared by one pass with a rotary tiller followed by one pass with a rotary harrow. We retrieved the soil for all experimental columns from this field on October 11, 2021. A total of 45 hollow cylinders, constructed of a perforated aluminum wall (24 perforations with a diameter of 2 cm) with an inner diameter of 10 cm, height of 9 cm and wall thickness of 4 mm, were used for the soil incubation experiments (Supplementary Fig. S10, SI). These columns were divided into three subsets of 15 replicates each: one subset served as the control group with no MPs addition, while the other two subsets included the incorporation of either indium-doped PET fibers or PET fragments into the top 2 cm of soil. The columns were uniformly packed with loose soil, well-watered to foster soil cohesion, and placed into the field. We then left the columns to drain over a 5-day period to further increase soil structural stability. Each column was covered with a dish to protect against potential heavy rain events during this time. Following soil compaction, the columns were carefully excavated and transferred to the laboratory to add the MPs spiked layer. To ensure consistent spiking depth, the soil within the topmost 2 cm of the aluminum cylinders was removed to make space for an MPs-spiked soil layer. For this purpose, 130 grams of sieved dry soil was mixed with 1 wt.% MPs (equivalent to 1.3 g) in Schott bottles. The mixture was homogenized using a Turbula® mixer for 5 min. The 1 wt.% MPs concentration was chosen to represent a higher contamination scenario, reflecting potential hotspots in areas with significant plastic pollution, while still being within the range observed in some industrially impacted regions. This level helps assess the impacts of elevated plastic concentrations in environments with intense agricultural or urban plastic use and was also selected to ensure concentrations were high enough to be reliably measured in the soil matrix by our method (ICP-MS). Subsequently, the homogeneous mixture was carefully poured into a metal ring with the same dimensions as the column, and 35 mL of water was added to maintain cohesion without compromising homogeneity. To enhance reproducibility of the spiked MPs layer, the wet soil-MPs mixture was compressed within the metal ring and delicately placed atop the columns. The control columns were processed using the same procedure, but without incorporating the MPs. All columns were stored in a dark cold room at 4 °C prior to transportation and X-ray scanning.

### Experimental design and schedule

Samples were kept in the dark cold room at 4 °C until X-ray scans were conducted (timepoint: year 0). On October 28th, 2021, the soil columns were reinserted into the field in a nested random distribution (Supplementary Fig. S9). Winter wheat was sown on the entire plot, including the columns. On November 3rd, 2021, all wheat seedlings in excess of three per column were manually removed. We harvested the winter wheat on July 11th, 2022. The wheat grown on the soil columns was harvested by hand, leaving a stubble of roughly 1 cm, while the wheat from the rest of the field was harvested using a bar mower, after the columns had already been removed from the field. All plants that had matured on a soil column were dried for further analysis to assess the impacts of MPs on the crop yield. On August 9th, 2022, the columns were retrieved and subjected to X-ray scanning for the second time (timepoint: 1 year). Subsequently, five columns of each MPs treatment (fibers and fragments) were randomly selected and sacrificed for further analysis to assess the vertical transport of MPs. Due to the destructive nature of the vertical transport analysis, the sample matrix was reduced to 10

columns containing MP fibers, 10 columns containing MP fragments, and 15 control columns. After two passes with a rotary harrow, all remaining 35 columns were reinserted into the field for continued incubation on August 24th, 2022. Winter barley was sown on the field and columns on October 3rd, 2022, where, once more, only three germinated seeds per column were retained. Barley was harvested on June 22nd, 2023, using the same method as for the wheat, and all plant material collected from the soil columns was dried. On October 12, 2023, all remaining columns were excavated and transferred to the laboratory for a final X-ray scan (timepoint: 2 years). Those columns containing fragments and fibers were sectioned for further analysis of the vertical transport of MPs. Supplementary Fig. S10 provides photographs of the column arrangement, field site, crop progression, and the appearance of a column post one year. Whenever the columns were removed from the field for analysis, they were wrapped in aluminum foil and stored in a dark cold room at 4 °C. The experimental field was fertilized according to agronomical best practice throughout the duration of the experiment.

### Sample digestion and ICP-MS analysis to assess MPs concentration

All soil columns were stored and maintained at −20 °C. The soil columns containing MPs ($n = 5$ for both fibers and fragments with wheat in year 1, and $n = 10$ for both fibers and fragments with barley in year 2) were carefully removed from the aluminum columns and sectioned into five layers using a stone saw, each approximately 2 cm thick, except for the bottom layer, which was variable between 0 and 1 cm due to uneven compaction over the experimental time. The layers were oven dried at 95 °C for 48 h, ground using a pestle and mortar, and sieved through a 1 mm pore size sieve to break up large aggregates. To prevent cross-contamination of MPs between soil layers during analysis, the lab equipment was rinsed with water and ethanol and dried after processing each layer. We began processing with the lowest layer (layer 5) and worked sequentially to the top layer, where the highest concentrations of MPs and indium were expected. The sieved soil was thoroughly mixed with water (~40 wt.%) for three minutes using a spatula to ensure a uniform distribution of MPs. Subsequently, three replicates (3 g) of each layer were weighed into a glass digestion tube (48 mL, from MWS GmbH), 15 mL of HNO$_3$ (65%) was added and subjected to microwave acid digestion using the Turbowave Simultaneous Automated Microwave Digestion System by MLS GmbH. The digestion process included: 1) the system pressure was gradually increased, reaching 60 bar over 20 min, 2) temperature was raised from 25 °C to 100 °C within a rapid 8-minute interval and maintained at 100 °C for 5 min, 3) temperature was further elevated to 250 °C over 15-minutes and maintained at 250 °C for 30 min, 4) temperature and pressure were returned to ambient conditions. Following sample digestion, the glass tubes were rinsed with Milli-Q water into polypropylene Falcon tubes and diluted to a final volume 50 mL. To achieve appropriate metal concentrations for subsequent ICP-MS analysis (7900, Agilent Technologies), a further sample dilution of 1:40 was performed. The ICP-MS system featured an integrated autosampler system (Agilent, SPS 4), sea spray Nebulizer, and nickel cones. Instrument calibration was achieved through the daily preparation of an indium (In) standard solution at a concentration of 100 μg L$^{-1}$, followed by additional dilutions to create calibration standards including of 0, 0.1, 0.5, 1, 2.5, 5, 12.5, and 25 μg L$^{-1}$. Throughout the analysis, Rhodium and Palladium (1 μg L$^{-1}$) served as internal standards. General quality assessment measures, i.e., procedural blanks and quality controls, were routinely included. MPs concentrations were then derived from the measured In concentrations using the experimentally defined In-to-plastic ratio (0.191 ± 0.004 wt.% for PET fragments and 0.185 ± 0.003 wt.% for PET fibers). The method detection limit (MDL) was determined to be 0.071 μg In L$^{-1}$. Translated into MPs concentration, this equated to a MDL of 1.52 mg MPs per 100 g soil. After obtaining the concentrations of MPs for each of the three technical replicates per layer, their values were averaged to yield a single mean concentration of MPs per layer for subsequent analysis.

## MPs recovery protocol and stability of indium tracer

The performance of the entire sample digestion workflow (including grinding, sieving, digestion and ICP-MS measurement) was assessed via spiked addition of PET fragments and fibers to additional soil samples collected from the experimental plot. Three replicates each of fibers and fragments (1 wt.%) mixed into soil were prepared. The MPs/soil mix underwent four wetting and drying cycles in an oven at 65 °C over a two-week period to induce soil settling and aggregation, thus more closely approximating soil at field conditions. After the last dry cycle, triplicates of each sample were weighed in the glass digestion tubes followed by acid digestion. Recovery across the entire analytical workflow was 96.2 ± 5.5% and 95.9 ± 6.7% for fibers and fragments, respectively.

To assess the possible extent of In leaching from the PET polymer matrix over the two year time span, we buried PET mesh bags (SEFAR PETEX®, mesh opening of 30 μm) filled with metal-doped MPs and sieved soil and placed them directly in the experimental field at a soil depth of 5 cm at the start of the experiment. The mesh bags remained in the soil for 2 years. A total of eight mesh bags were utilized, each containing either PET fragments or fibers (100 mg) alongside 2.4 g of sieved soil. Following the two-year exposure period, the mesh bags were retrieved, and the In content was analyzed using ICP-MS after acid digestion. The concentration of In per MPs remained nearly constant throughout the duration of the study, indicating the In could be used as a conservative tracer for MPs throughout the experiment. For PET fragments, the initial In concentration was measured at 0.198 ± 0.006 wt.%, which only slightly decreased to 0.191 ± 0.004 wt.% after two years in the field. Similarly, PET fibers had an initial In concentration of 0.186 ± 0.008 wt.%, with a concentration of 0.185 ± 0.003 wt.% after the two-year exposure period.

## Crop yield analysis of wheat and barley

The hand-picked wheat and barley crops from the columns were dried at room temperature for a minimum of four weeks. Following drying, the grains were separated from the remaining plant material. Total biomass, the grain weight, and the weight of the residual plant material (e.g., straw) was measured. The harvest index was calculated as the ratio of grain weight to total biomass. The results suggest that the addition of MPs had very little to no effect on the crop yields of wheat and barley overall. Although the presence of MPs fibers likely results in a slight reduction in total barley biomass and grain weight compared to the control (estimate total biomass = −10.2 g, $p = 0.051$ and estimate grain weight = −4.28 g, $p = 0.0625$). All results from crop yield analysis are presented in Supplementary Fig. S12 in the supplementary section and the statistical analysis in Supplementary Section S11 (SI).

## X-ray CT image acquisition

X-ray images for the initial time point (year 0) were acquired using YXLON, FXT-225.48-3, 225 kV X-ray tube and a YXLON XRD 1621 AN18 ES detector at Eurofins Qualitech AG. The measurements at year 0 were performed with 15 replicates for fiber columns, 15 for fragment columns, and 15 for control columns. The 3-D X-ray images were reconstructed from 1710 projections and had a voxel edge length of 99 μm. The X-ray data for the 1- and 2-year timepoints were acquired using a GE Phoenix v|tome|x 240 industrial X-ray scanner equipped with a detector GE DRX250 detector. For each 3-D image, 2000 projections were obtained. At year 1, the measurements were performed with the same replication as year 0: fragments ($n = 15$), fibers ($n = 15$), and controls ($n = 15$). At year 2, the measurements included fragments ($n = 10$), fibers ($n = 10$), and controls ($n = 15$). The voxel edge length was adjusted to 114.8 μm. The X-ray energy level used for image acquisition were adapted to the field-moist bulk density of the soil sample. Scanner setup and reconstruction parameters are provided in Supplementary Table S1.

## X-ray CT image processing

ImageJ/Fiji[9,75–78] with the SoilJ plugin[79] was used to process and evaluate the three-dimensional X-ray images. Initially, all images from year 0 were scaled to the same resolution as in year 1 and year 2. Subsequently, the soil column outlines within each image were automatically detected using SoilJ. The greyscale of all three-dimensional images was calibrated to the gray-value of the column wall (aluminum) and the 0.1 percentile of the gray-value, which was used as a reference value for air. The greyscale calibration process was applied on each horizontal image layer, which enabled the correction of image illumination biases in the vertical direction[79]. Joint histograms illustrating the mean distribution of the calibrated gray-values extracted from all three-dimensional X-ray images per year are shown in Supplementary Fig. S11. We then applied a median filter to all three-dimensional X-ray images with a radius of two voxels to reduce image noise, followed by an unsharp mask with standard deviation of two voxels and a weighting factor of 0.6 to sharpen phase boundaries in the images. To segment the three-dimensional X-ray images into soil matrix, air-filled pores and particulate organic matter (POM; including roots and soil macrofauna), Ilastik, an interactive machine learning tool for (bio)image analysis[80], was used. We developed three separate algorithms, one for each scanning timepoint, using a training dataset of 18 image stacks in total. Each timepoint had six image stacks, with each stack measuring $100 \times 100 \times 100$ voxels. We annotated these image stacks into three classes: soil matrix, air-filled pores, and POM. Based on our annotations into the three classes, a Random Forest classifier was trained for each year's dataset. This interactive process allowed us to create tailored algorithms that are specifically adapted to the unique characteristics of the data from each timepoint. Internally, the out-of-bag error of the Random Forest was 6.8%, 7.4% and 3.0% for the three respective scanning timepoints. After the classifier was trained, it was applied to all the median-filtered and unsharpened X-ray CT images in full image resolution to segment them into the three defined classes. We estimated the image resolution to be 2–3 times as large as the voxel edge length, i.e. approximately 300 μm, which is sometimes used as a threshold diameter to define soil macropores[81]. We therefore referred to all imaged pores as macropores hereafter.

## X-ray CT analysis of soil morphology

The morphology of the macropore space was examined in three distinct regions of interest (ROIs). ROI 1 encompassed the largest common region found in all years (height = 65 mm, diameter = 75 mm). The diameter of ROI 1 was 15 mm smaller than the inner radius of the aluminum cylinders to account for the artificial gap that forms between the soil core and aluminum due to wet/dry cycles or even freeze/thaw cycles, creating air-filled spaces. ROI 2 consisted of the region where we anticipated the highest concentration of MPs, that is, where MPs were initially mixed into the sieved soil at the top of the column (height = 10 mm, diameter = 75 mm). ROI 3 corresponded to the region where we expected less MPs accumulation at deeper column positions, starting from a depth of 35 mm (height = 40 mm, diameter = 75 mm). Due to the compaction of soil over the 2-year time span and the deposition of new soil on top of the columns, we had to visually adjust the starting positions of ROI 1 and ROI 2 to align with the vertical position of the sieved soil layer. SoilJ[79] (version 1.8.0) was used in combination with MorphoLibJ[82] to analyze the characteristics of the imaged macropore network, biopores and POM. Our analysis of the imaged macropore network encompassed a range of morphological measures, namely soil macroporosity, specific surface area, critical pore diameter, and a unitless local connectivity measure, the connection probability[83]. Macroporosity is defined as the ratio of the macropore volume (void spaces) to the total bulk volume of the soil and the specific surface area of macropores is the total surface area of the macropores per unit volume of soil. The critical pore diameter corresponds to the bottleneck in the pore connection from top to bottom surface of a ROI. Its squared value is known to be correlated with the saturated hydraulic conductivity of undisturbed soil[84]. Additionally, we computed the pore size distribution using the maximum inscribed sphere method ('Local Thickness') as implemented in SoilJ and the distance of a soil matrix voxel to the nearest top-surface-connected macropore, the latter providing a proxy for soil aeration[85]. We delineated biopores from the segmented imaged macropore networks using the algorithm published in Lucas et al.[86] as implemented into SoilJ. We focused on biopore volume fraction, specific surface area and pore size distribution of biopores. For the characteristics relating to POM, we only investigated the volume fraction and specific surface area of POM within the soil matrix.

## Statistical analysis

Statistical analysis was performed using RStudio 2022.07.1, with R version 4.4.2. The study aimed to analyze the effects of Replicate, ROI, Year on various morphological measures and the effect of Replicate, Layer, Year on vertical transport of MPs. The experimental design consisted of 15 Blocks, with each Block containing one plot for each replicate: control, MPs fragments, and MPs fibers. These treatments were nested within the Blocks. The analysis included multiple Replicates (control, MPs fragments, MPs fibers), ROIs (ROI 1, ROI 2, ROI 3), Years (Year 0, Year 1, Year 2) and Layers (Layer 1 - Layer 5). The effects of Replicate, ROI, and Year on various morphological measures were analyzed using linear mixed-effects models (LMMs). The models included fixed effects for Replicate, ROI, and Year, and random effects for Block and interactions between Block:Year and Block:ROI to account for variability within experimental blocks. Outliers were identified using the interquartile range (IQR) method and excluded if they fell outside 1.5 times the IQR, provided they were confirmed as true outliers and not biologically relevant values. Post-hoc pairwise comparisons were conducted using the emmeans package to assess differences in Replicate across each ROI and Year, as well as temporal changes across Year within each Replicate and ROI. The p-values were adjusted using Tukey's HSD method for multiple comparisons. The results of the models and post-hoc comparisons are summarized in Supplementary Section S11 (SI) for each morphological measure in Supplementary Tables S2.1–S10.2. We used a separate linear mixed-effects model for the MPs vertical transport analysis to assess the effects of Replicate, Layer, and Year on the vertical transport data. The model included fixed effects for Replicate, Layer, and Year, as well as a random effect for the interaction between Block and Replicate. The log transformation of the vertical transport data was applied to handle data skewness. Outliers were identified and excluded using the same IQR method as described above. Post-hoc pairwise comparisons were conducted for Replicate across each Year and Layer, as well as for Year across Replicate and Layer and Layer across Replicate and Year, using the emmeans package. The p-values were adjusted using Tukey's HSD method for multiple comparisons. The results of the vertical transport analysis are shown in Supplementary Section S11 (SI) in Supplementary Tables S11.1–S11.3. To assess the effect of Replicate and Material on the crop yield data, a separate linear mixed-effects model was used. The model included fixed effects for Replicate and Material, as well as random effects for Block and the interaction between Block and Material. Outliers were identified and excluded using the same interquartile range (IQR) method as described above. Post-hoc pairwise comparisons were conducted for Replicate within each Material, as well as for Material across Replicate, using the emmeans package. The p-values were adjusted for multiple comparisons using Tukey's HSD method. The results of the crop yield analysis are detailed in Supplementary Section S11 (SI) in Supplementary Tables S12.1–S12.3. In all figures, statistical significance is indicated as follows: $p$-value $< 0.001$ "***", $<0.01$ "**", and $<0.05$ "*".

## Supporting information

Eleven figures and three tables, including further information on: Temporal evolution of soil morphologies in ROI 3 (Supplementary Fig. S1), impact of MPs on soil morphologies in ROI 1 (Supplementary Figs. S2 and S3), impact of MPs on soil morphologies in ROI 2 (Supplementary Figs. S4 and S5), impact of MPs on soil morphologies in ROI 3 (Supplementary Figs. S6 and S7), temperature and precipitation data (Supplementary Fig. S8), random column distribution in the field (Supplementary Fig. S9), visual insights of the field site (Supplementary Fig. S10), X-ray scanner setup and reconstruction parameters (Supplementary Table S1), histogram of calibrated gray values from X-ray images of different years (Supplementary Fig. S11), impact of MPs on crop yield (Supplementary Fig. S12), statistical analysis of soil morphologies (Supplementary Tables S2.1–S10.2), statistical analysis of vertical transport of MPs (Supplementary Tables S11.1–S11.3), statistical analysis of effect of MPs on crop yield (Supplementary Tables S12.1–S12.3).

## Data availability

The dataset for plots of the manuscript and statistical tests are available on figshare https://doi.org/10.6084/m9.figshare.28555058.

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

## Acknowledgements

Roman B. Schefer and Denise M. Mitrano were funded through the Swiss National Science Foundation (grant number PCEFP2_186856). We thank Thomas Bucheli for advice on conducting the field experiments at Agroscope as well as Dani Fuchs and the field crew for preparing and harvesting the field site. We also extend our appreciation to Mélanie Emery for her assistance in analyzing aspects of the vertical transport of MPs during her master's project. Additionally, we appreciate the assistance and access provided by Ana Lorena Abila and Claudio Madonna from the Geological Institute at ETH Zürich, who facilitated the use of the stone saw.

## Author contributions

R.B.S. executed experimental work and data analysis. R.B.S., J.K. and D.M.M. evaluated data. R.B.S., J.K., and D.M.M. discussed and interpreted data. All authors were involved in experimental design. R.B.S. and D.M.M. wrote the manuscript, with input from J.K. All authors approved the final version of the manuscript. D.M.M. and J.K. conceived of the project. D.M.M. supervised and acquired funding.

## Competing interests

The authors declare no competing interests.
