## [Transparent Peer Review file · Communications Earth & Environment]

Minimal vertical transport of microplastics in soil over two years with little impact of plastics on soil macropore networks

Corresponding Author: Professor Denise Mitrano

Version 0:

Decision Letter:

Dear Professor Mitrano,

Your manuscript titled "A Two-Year Incubation Study: Investigating the Vertical Transport of Microplastics in Soil and its Impact on Soil Pore Development" has now been seen by 3 reviewers, whose comments are appended below. You will see that they find your work of some potential interest. However, they have raised quite substantial concerns that must be addressed. In light of these comments, we cannot accept the manuscript for publication, but would be interested in considering a revised version that fully addresses these serious concerns.

We hope you will find the reviewers' comments useful as you decide how to proceed. In addition, for publication in Communications Earth & Environment to be appropriate, we would need you to:

1. Present compelling new insights into the vertical transport of microplastics in soil and its impact on soil pore development.
2. Re-assess the statistical analysis associated with your experimental design, in light of the multiple factors and repeated measures. Provide comprehensive details of your statistical analyses.
3. Place the experimental plastics used in this study in a broader context of soil plastic pollution.

Should additional work allow you to address these editorial thresholds and all the reviewers' criticisms, we would be happy to look at a substantially revised manuscript. If you choose to take up this option, please either highlight all changes in the manuscript text file, or provide a list of the changes to the manuscript with your responses to the reviewers. In addition, please describe how you have addressed the 3 editorial thresholds.

If the revision process takes significantly longer than three months, we will be happy to reconsider your paper at a later date, as long as nothing similar has been accepted for publication at Communications Earth & Environment or published elsewhere in the meantime.

Please use the following link to submit your revised manuscript, point-by-point response to the reviewers' comments with a list of your changes to the manuscript text (which should be in a separate document to any cover letter), a tracked-changes version of the manuscript (as a PDF file) and any completed checklist:

Link Redacted

**** This url links to your confidential home page and associated information about manuscripts you may have submitted or be**

reviewing for us. If you wish to forward this email to co-authors, please delete the link to your homepage first **

Please do not hesitate to contact us if you have any questions or would like to discuss the required revisions further. Thank you for the opportunity to review your work.

Best regards,

Kate Buckeridge, PhD
Editorial Board Member
Communications Earth & Environment
orcid.org/0000-0002-3267-4216

Somaparna Ghosh, PhD
Associate Editor
Communications Earth & Environment

EDITORIAL POLICIES AND FORMAT

If you decide to resubmit your paper, please ensure that your manuscript complies with our editorial policies and complete and upload the checklist below as a Related Manuscript file type with the revised article:

Editorial Policy Policy requirements
(Download the link to your computer as a PDF.)

- Behavioural and social science
- Ecological, evolutionary & environmental sciences
- Life sciences

<https://www.nature.com/documents/nr-reporting-summary.zip>

For your information, you can find some guidance regarding format requirements summarized on the following checklist: (<https://www.nature.com/documents/commsj-phys-style-formatting-checklist-article.pdf>) and formatting guide (<https://www.nature.com/documents/commsj-phys-style-formatting-guide-accept.pdf>).

REVIEWER COMMENTS:

Reviewer #1 (Remarks to the Author):

This is a thoroughly conducted study on a very important topic, described in an excellently written manuscript. The authors use highlight sophisticated advanced X-ray CT tools to provide an in-depth assessment of micro-plastic addition effects on soil structure. The study is of great interest and importance for better understanding of the impacts of plastic pollution on the environment. Thus, I strongly recommend publication of the manuscript.

Suggestions for improvement:

- 1) One major suggestion for improvement that I have is related to statistical analyses conducted in the study. The experiment has a very solid design with plenty of replications. It appears to be a 2-way factorial with factors: type of MP addition and the year of the observations. Plus, for some of the studied variables, e.g. transport related variable, the soil depth appears to be the third studied factor. Given all that, I am not sure I follow the reasons for the type of analysis (one-way ANOVAs) conducted here and whether it is really appropriate for this study. The effects of all these studied factors are either directly or indirectly mentioned and reported in the Results and involved in the Discussion, yet, the conducted one-way ANOVAs really cannot provide adequate statistical support for those. Why not do a proper 2-way factorial split-plot (or, better, repeated measures) analysis to address significance of the treatment and year factors and their interactions? Or 3-way split-split-plot analysis with depths as a repeated measures? That would greatly strengthen the study! One other related question I have is the choice of the non-parametric tools for the analysis – from the graphs (e.g., Fig.4) it looks like often it might be just one or two extreme outliers that drastically increase variability and lead to the lack of statistical significance. I would recommend exploring either excluding these outliers or transforming the data prior to the analyses. I believe that might generate results with a greater statistical power.
- 2) It would strengthen the relevance and importance of study to provide a justification of how the plastic materials created here are relevant for typical plastic pollutants found in the soil.
- 3) The separation into the three regions of interest (ROI1, 2, and 3) makes perfect sense, however, having these regions described only in the Methods makes it impossible to interpret the Results (which come prior to the Methods). I would recommend adding a brief description of these three regions early in the Results. Also, it might be helpful for the reader is using more meaningful names for them - e.g., "entire core" or such for ROI1, top of the core... instead of just umbered ROIs

Minor questions/comments:

L.148 – What ROI is shown on Fig. 3? Seems like a discrepancy between the text and figure caption.

Fig. 2 – Please specify the lower boundary of macropore size. It is not zero, right?

I.173 Fig. 3 - Would not "Visible on the images" be clearer? Also, my understanding is that time zero scans are for initially packed cores – how come they already have so many biopores. Are these definitely biopores?

I.204 – Is this correct that here "transport" is actually the concentration of MP found at different depths? If so, would be better to say so.

Fig. 5 – Marking and mentioning statistical significance in terms of treatments, years, and depths would be beneficial here.

I.487. Please provide formal definitions and determination methods for microporosity and specific surface area and I. 494 for pore size distribution.

I.503-504 What is the purpose of determining "significant differences between replicates"? Or is it just incorrect description?

Reviewer #2 (Remarks to the Author):

The transport of plastics in soil, particularly in agricultural systems, is an important issue. While much research has been conducted over the past decades, the paper by Schefer et al. makes a useful contribution by employing labeled plastics to enhance our understanding. I recommend the manuscript for publication, but only after major revisions. While the results are largely predictable, the use of labeled plastics represents a valuable methodological advance.

Abstract: The assertion that plastics are primarily used unintentionally in agriculture is inaccurate. Please address and expand upon this misconception in detail within the revised paper. The main use is intentional.

Line 19: Remove the word "significant" as it reflects incorrect statistical interpretation of p-values.

Line 21: Clarify that microplastic transport was very limited, with observations restricted to the top few centimeters, and only 1% reaching deeper layers. This crucial finding from the results must be included in the abstract.

Line 25: The statement overstates the paper's relevance. The manuscript focuses on plastic transport, soil structure, and macropore evolution, but these are just a few aspects of plastics' broader impact on soil. You do not, for example, address the release of additives or effects on plant productivity and soil microbiomes. Please moderate the claim to reflect the paper's specific focus. Your conclusion handles this well—consider adopting a similar tone here.

Line 34: Consider removing normative jargon such as "mindfulness." The term is overused and detracts from the scientific tone.

Line 43: Avoid terms like "pivotal." Soil fertility has many contributing factors, and plastics are just one aspect. The paper tends to overemphasize its findings—please moderate these claims and use a more neutral, matter-of-fact tone.

Line 51: The point about synthetic origin is unclear and needs precision. What are these "important questions"? Why is the synthetic nature of plastics the main issue? Would biobased materials present less of a problem? Please clarify these points, as the current phrasing is vague.

Line 84: There is a substantial body of research on particle transport in soil. While you cite a few relevant papers, the existing literature on older colloidal transport (e.g., Kretschmar, Ryan, Elimelech, and others) is more comprehensive and should be referenced. The claim that nanoplastics would be more mobile is incorrect; according to filtration theory, particles around 1 μm in size are most mobile. Smaller particles, like nanoplastics, are likely less mobile. Please revise this statement.

Line 98: The term "unravel" overstates what can be achieved. You are contributing to the understanding of certain aspects, not resolving all complexities. Please temper this statement to be more realistic and precise.

Line 163: I disagree with the statistical analysis. Please see my comment on Line 497 for further details.

Line 201 and following: The finding that microplastics are relatively immobile in soil is consistent with theoretical predictions from prior research. I appreciate the innovative approach of using labeled microplastics to confirm this.

Line 497: I strongly disagree with the use of p-values to determine statistical significance. The concept of statistical significance based on p-values has been extensively critiqued. Please refer to Amrhein et al. (Nature, 2019) and the ASA's guidelines on this issue. It is incorrect to assert that a p-value of 0.05 defines significance. I recommend fully revising the statistical analysis in accordance with current guidelines. Compare:

<https://www.nature.com/articles/d41586-019-00857-9>

and

<https://reproductive-health-journal.biomedcentral.com/articles/10.1186/s12978-021-01131-w>

Reviewer #3 (Remarks to the Author):

A two-Year incubation study: investigating the vertical transport of microplastics in soil and its impact on soil pore development.

Synopsis

This manuscript reports the findings from a field study, where cores containing soil spiked with PET microplastics which were doped with indium were exposed to an agricultural setting growing wheat and barley. The study employed state of the art techniques to track where microplastics ended up in the soil profile of the cores, but also explored the soil porosity and biological aspects thereof at three time points. The study reports that there were minimal effects overall when comparing the treatments to the controls, but over time the pore structure was different in the controls. Furthermore, levels of indium as a proxy for microplastics were found in deeper soil layers, particularly over time, but perhaps not as much as the authors expected which suggests that the majority remained in the initially spiked top layer. The authors discuss the findings in light of biological and physical processes, suggesting that root growth but also possible burrowing by animals may explain this observation. The authors conclude that sustainable alternatives [to conventional plastic types] should be explored to reduce the effects of microplastics in soils.

I really enjoyed reading this manuscript, it is presented at a very high quality, which would be expected. The amount of effort which has gone into this piece of research is impressive! The research questions are interesting and will be of great interest to researchers in the discipline, particularly how the authors have employed ways to explore soil structure and pores in situ (which is not that new in itself, but a great way to do see what's inside). The indium tracer technique used appears to have been employed successfully by the research group previously and the authors report that there are appropriate quality controls in place. There are however some points which are concerning me; they are not necessarily related to the methods and associated analytical aspects, but more so on the wider implications, the experimental design and implementation, statistical analysis and the presentation thereof, and the quality of the discussion. I have laid out the summarising points in the main comments below, and I took the liberty to provide more in-depth comments and suggestions to provide more details to support the main comments, but also some points which could be clarified more in my opinion.

Major comments

- The statistical analysis needs much more transparency to show the readers what was done exactly. The methods section suggests that the number of replication changed over time, which is acceptable, but you will have to show how that affected the degrees of freedom (for the ANOVA) and thus the calculated p-values upon which conclusions are based. P-values on their own are rather uninformative and murky at best. Furthermore, there are subsamples as well to measure response variables, probably as technical replicates, but it is not clear what you did with those to assess the measurements at the biological level. For example, with the layers included, readers could interpret the methods that up to 300 ICP-MS measurements were done, but this is quite unclear. Throughout the methods it would be imperative that all replication is clearly explained.
- The methods also state that a nested field design was employed, supported by figure S9 (please check this figure for the removed cores!). These are standard agronomic field tests, nothing wrong with them, but they are nested to account for random spatial variation. Some call them "blocks", yet there is no clear consideration given in the statistical analysis for this nestedness. It will affect how the degrees of freedom and which sums of squares are used for the parametric test. This also brings me to the point of non independence of time points, the cores, while removed in year one, essentially remain the same (bar from removing $n = 5$ from the treatments), and are put back into the soil. Whatever was there in $t = 0$ probably affects $t = 1$, and that is carried over to $t = 2$. The statistical analysis ideally should take this into account, but that is not made clear and therefore I suspect it was not considered. It will probably require consideration and could lead to a change in the significance of some results.
- The cores included three very big plants (monocots) which of course would have a major influence in producing pores due to their rooting. While this does not invalidate the findings, the interpretation thereof needs care, because there study design does not exclusively test the effects of microplastics on pore formation, instead, as a whole, it tests any direct and indirect effects (on the plants). I think that reporting non-significant results are just as important as significant ones, but they still need to be contextualised just as carefully. If the aim was to simulate an agricultural setting, with all relevant aspects for as much feasible, then that has been achieved – the unravelling of interactions, however, particularly mechanistic ones, are not that clear however. For that a set of unplanted cores would have been needed, which of course would double the amount of work.
- Were there any soil animals present in the cores? I know it was not part of the aim, but could be interesting to explain some findings due to homogenisation (worms are particularly good at that).
- Technically, the effects of microplastic fibres and fragments WITH indium was assessed, not microplastics without the added metal, how little it may be. I completely understand that the idea was to track the plastics using indium as a proxy for the plastics assuming no leaching occurred from the plastics (which you have controlled for, that is great!), but this additive could be considered as an extra aspect in the design. The plastics are obviously model particles, because according to the methods they have been processed after purchase and modified to fit the purpose of the study, therefore it should be noted that these particular model plastics are not necessarily reflective of what is typically found in an agricultural soil. This aspect does not invalidate the results, but it does require acknowledging the translatability of such models to a wider context even if it was carried out in a more natural setting than a controlled lab study otherwise. It could have been done with plastics which were not metal-doped, but of course then you have the issue with not being able to track them (but can still assess the pore formation). I suppose this would be a "procedural control" for the manipulation of the experimental units, which are the plastics.
- I understand that the crop was harvested too, and used to assess yield (see methods), but the results are not presented here. I do understand that this was not part of this particular project's questions, but in light of the claims related to

extrapolation to field settings, particularly with the aim in mind of informing “decision makers”, this finding would be of interest.

- Eventhough the experiment encompassed an impressive two years AND was done in the field, the experimental units were still rather confined, with the cores being 10 cm diameter and 9 cm deep – particularly when realising that each core supported three plants and their rooting system for two years. I think this aspect could be considered when discussing the findings as well.

Minor comments and suggestions for change

Title

Informative, but please check that it clearly conveys the message: with “its” do you refer to vertical transport (singular noun), or microplastics (plural noun), or the impact of “investigating” as a whole? I know this is rather pedantic, but I also think it is important to avoid confusion with the first thing someone will read.

Keywords

I wonder if “metal-doped plastics” is probably not a string of keywords someone will use when searching for this particular type of research. Just a thought.

Abstract

line 12: why only “unintentional”, could be any type of introduction, intentional or not?

Line 13: perhaps remove “fertility” given that this is not the focus of the study (fertility *sensu stricto* was not measured as part of the study)?

line 21: were the pores in the controls different from the treatments, or were the significant changes over time (or indeed, both)? Can this be clarified in the abstract given what is stated in line 23?

line 25: what is the take-home message (impact) from the study, other than this allows for determinations of realistic transport rates on soil? I would appreciate a more impactful concluding statement.

Introduction

Line 28-31: perhaps this sections about the UN-SDG is not relevant for the nature of the study, I appreciate it is *important*, but it may not be needed to introduce this highly focusses study. Perhaps jump straight into the issue of soil structure and how there is evidence of how microplastics may affect that?

Line 43: Not a huge issue, but agronomists maybe disagree that fertility *sensu stricto* is not necessarily soil structure, nor vice versa. Of course, soil structure is important to make a soil more accessible to resources. I would be careful with this and the wide readership in mind of the journal.

Line 48: are you sure that “evolution” would be a correct term (given the traditional connotation with the biological adaptation to biotic and abiotic parameters), or could we think of this as ‘development’?

Line 51: why is it the “synthetic origin” that raises these questions?

Line 54: what are the “qualitative” effects these sources [21-24] demonstrated on “soil systems”, can you give an example? Why specifically “qualitative”?

Line 56: I see that you try to justify the novelty of a long-term field study here, but since these sources [21-24] much more has been done (there are a lot of scientific studies on this topic now), including field-based work which has appeared recently. This could be made much stronger by also emphasising newer studies to show a wider picture of what has been done in this regard. In fact, you cited [27] later on for this?

Line 60-61: “quantifying MPs transport through soil depth over time will help constrain risk assessments by better understanding temporal exposure.” is quite vague, at least I had to read it a few times to get a sense of what you actually mean: what are risk assessments and how are they conducted with the added knowledge? What is temporal exposure – is there any other type of exposure we do not understand that well? I think I know what you mean but for clarity, this could be reworded.

Line 64: “easily” is subjective. I bet many people will not think it is an easy thing to do.

Line 96: Has this technique been used before with similar particles? If so can you add another source to the establishment of the method, if not, then we will be curious to see the quality controls in your study (which should be there anyway, but even more relevant for method validation).

Line 98: I don't think you have really explored the predicted or possible “interactions” that much to unravel – what exactly do you refer to with *interactions* for this study (that they somehow cause more pores? Bigger pores, smaller pores? That plastics go through pores become adhered to pore walls?)

Line 99: can you explain how unravelling these expected complex interactions will lead to informed decision making? What

will this information change for making any current decision? While I understand that impact is important, statements such as these will need to be supported and clarified. Almost everyone can say that their research will 'inform decision making', but not many follow through which type of decisions may be affected and by who will make the decision (farmers? "the public"? local government? UN?). Banning plastics altogether, because they have (complex) interactions with soil pores? Swapping for biodegradable plastics because... I know this may come across perhaps a bit harsh, but I think it is important to consider the implications of such statements!

Line 102: what is an "extended two-year timeframe" is that with "extended" more than two years? If so, can you be specific please? Could you just say "over two years"?

Line 102: are elements always with a capital letter when spelt in full (I know they are in the periodic table)?

Line 103: what is a "packed" soil column – not everyone will know it? Incubation would suggest a highly controlled situation, like in an incubation chamber. I would use a different term. "Exposed", perhaps? What was the agricultural field: arable with cereal crops (wheat and barley) over the two years?

Line 106: periodically = after one and two years?

Results

Thank you for presenting summary results as boxplots; they are perhaps more informative than means with errors, BUT to enhance and work towards complete transparency, the raw data are easily transposed on the same graphs. It would be good to start seeing a movement in the scientific community to present the spread (distribution) of the data so that there is no inadvertent hiding of data behind the central tendencies and summarising spread.

From the methods, technical replicates were sampled ($n = 3$ from layers) but it is not clear what was done with those to calculate the biological replicates.

I checked Table S9 and see comments below for the statistical analysis section in the methods about reporting statistical results more transparently. It seems you compared the years, which in an interesting question. However the cores went back into the field each time and are not independent (paired), I would assume. For example, any existing pores in year 0 or 1 would have an effect on the subsequent year(s) whether they remain or disappear – also the df would have changed over time because pots were removed, but only for the treatments. P values of 0 logically not meaningful (absolutely no chance at all), therefore, especially with the sample sizes used, please report as $p < 0.001$ where relevant, but not as 0.000.

I also checked Table S10. It appears that the significance (based on the reported p-values) are mostly at $T = 0$, so straight after the cores were prepared as far as we can tell from the methods. How come that there are significant differences before deployment for a variable which is expected to be the same at $t = 0$ (in macropores in ROI2). If there are inherent differences of response variables between designated experimental units at day dot, then how will that influence subsequent observations?

Why is figure 2 highlighted to show only the results from the controls and not reflecting the research aims involving the comparative nature of the experimental treatments?

I wonder if you could see the pore distribution as multivariate data e.g. the results in line 133 - 137, and thus approach this with a multivariate mind (the pore size profiles can be compared for example, rather than a summarising univariate measure or single size classes). Could be done with an ordination, or permutational techniques. Just a suggestion as it may test hypotheses relevant to changes in pore sizes, but then as a whole.

Line 129: I was not aware that the aim was to simulate a freshly prepared seedbed, but also the methods do not really simulate a freshly prepared seedbed (I think that is done differently usually, especially in the field).

Line 128 - 143: Is this section describing results for ROI3 in the controls only, given that this refers mostly to figure 2? It is not clear why there is such an emphasis on this bit of the controls only which is not really reflecting the aim of unravelling "complex interactions" of microplastics that well.

Is Figure 3 also the controls only? I assume these are some key examples of the images and do not represent the full set of results?

Line 202 and 205: is it also notable when the top layer contained all the MPs to begin with? Also, obviously it was the "highest" because it was the *top* layer; "greatest" or "most" maybe?

Line 206 - 208: this is discussing the results, and it probably should go to the discussion section where you can focus on explaining these findings more, supported by scientific literature.

Line 210: are you referring to all layers you sampled for the observation that the LOD was exceeded?

Line 212 Would you agree that losing from the system about 16% of the fragments you added to the system is maybe something to worry about? Quick maths (please double check): you added 1.3 gram of the plastics to a core (see line 362), so at a recovery ratio of 83.4% (the lowest recovery, after 2 years) there was a loss of $0.16 \times 1.3 = 0.208$ gram. Scaling that up

it would be 26 gram m⁻² of fibres and / or plastics, which I would argue is quite substantial. Is this a technical loss due to measurements and rounding (but within the accepted error margins), or can you find out that these particles did not end up somewhere else in the field?

Figure 5: I wonder if you can place these depth profiles side by side based on year? It looks like the x-axis is log scale, so that distorts the boxes (IQR) and the whiskers. Can you also state this in the caption?

Discussion

The first part of the discussion (221-242) does not seem to reflect much on the overall aims of the study regarding interactions with microplastics. Instead, this section covers mostly general soil physics, which of course is important as a foundation, but could be brought in light of the experiment and its setting more.

Line 221: Be careful not to cherry-pick the significant findings only. However, why can't we explore this finding more that the MPs are not causing any pore changes (which is for better or worse) I would argue any deviation from the control is something to explore further, even if it is not significant internally for a treatment.

Line 231: was there any evidence that there were earthworms in your soils (and more importantly the cores)? Otherwise this explanation, why valid in general, does not hold well for the observed results. Furthermore, the cores did contain three plants with their roots, including after being harvested and (I assume) left to decompose.

Line 244 – 262: it is quite curious to see a large section in the discussion such as this with no support from the scientific literature – unless I missed something, I see no citations at all here. Please add relevant citations to this section, particularly to key statements which certainly need it (e.g. line 253, but there are many more, particularly line 256 where you curiously mention leached metals, which you obviously added to the plastics as well...). Without proper support from relevant scientific evidence, this will be considered poor form I am afraid.

Line 245: I don't think you explicitly measured burrowing nor its behaviour, at least it was not clear from the methods. The results only show pores at different time points, but it is an assumption that these are caused by burrowing. Please be careful with interpretations like this.

Line 246: ah now I can see that you consider the spiking is considered high and there is something about highly contaminated sites. There is still no citation for this, nor much of a justification to add that level of MPs to an arable soil.

Line 264: the presence of indium in the soil was shown, which you can assume was representative of the microplastics (slight difference!).

Line 270: do you think that another similar experiment but without plants would help test part of that hypothesis? Highest = greatest or most?

Line 273: not sure where the nanoplastics come into play here, it seems to be a distraction.

Line 276 - 278: it is also worth considering that [39] employed *Lumbricus terrestris* a large worm, which has quite a distinct ecological niche in that it burrows deeply (up to metres deep), lives in permanent burrows and pulls in decaying material to ferment in the burrow. I always wonder how representative this species is for pot studies, even done in a field (but that is irrelevant now) – however, there are probably more processes in the soil than just a big burrower, and you would have seen evidence of such activity on the surface (I think it is likely that a single *L. terrestris* would have a complete 706 mL soil core for breakfast, if they were there). I would recommend using different evidence to support your hypothesis for this phenomenon.

Line 280 - 286: this explanation for the loss of MPs will also need support with relevant science, please.

Conclusion and implications

Line 288 – 293: this part is introductory and not based on the results, the conclusions and implications should be based on the findings presented in this study.

The rest of the conclusion section is largely a summary, instead it would be better to highlight the importance of the findings. The last part looks good about wider implications, but the use of alternative plastics as a recommendation needs much more research before they are to be deployed without testing and regulation.

Materials and Methods

Line 315: I checked (on the 26th of September 2024) the online catalogue of the company for this type of PET and I can't seem to find the particular product you refer to. I did find PET "yarns", which I suspect were used, but what else did these plastics contain other than PET – did you check? Can this be clarified, perhaps you already did so in your work presented in [51] and [52], or elsewhere?

Line 323-234: this is a bit confusing, are these sentences double per chance? Also I can see from source [51] that fibres were produced, and cut at around 500 um. What are the difference from the current methods compared to that described in your work presented in [51] and [52]? Careful with self-plagiarism, I know that is difficult for repeated methods though.

Line 351: how many 2 cm holes were there and were they covered somehow to keep soil inside (if needed)? Consider referring to FigS10?

The cores were about 706 mL, with a depth of 9 cm, which could be considered rather confined for relevant mesofauna such as earthworms (typical pore formers) and even plant roots. I understand that larger cores would significantly increase the cost, but the relatively small size of the physical experimental units should also be considered when extrapolating the results.

Line 362: you may have expected this question: what is the rationale to spike the top soil layer at 1% w/w it is quite... a lot? I understand that for the cores as a whole the density is less, but that was not done.

I know that mixing plastics homogeneously into soil, particularly fibres, can be very challenging, and fibres at 1% can be very 'fluffy'. However, the mechanical mixing of the plastics in the 2 cm layer would very likely have changed the pore composition and soil structure (and bulk density) compared to the rest which you let sit for 5 days to settle, even after "delicately" compression. Can you comment/clarify how the vertical continuation of the physical composition was ensured?

Line 371: "reinserted" - I understood that the samples were kept in the lab until this point?

Line 372: With such small surface area (~79 cm²) even three seedlings are quite high (382 seeds m⁻²). Can you comment on how this represents a typical sowing density? There are three big, maturing wheat plants growing in a 706 mL container (ok, it is in the soil) so how is that representative of 'the rest of the field', and then there will be barley the next year.

Did the wheat seeds germinate in 7 days and all of them were in the pots were viable? Were there any roots in the soil which you had to remove or left (these would be pores, albeit probably small)?

Line 372: Figure S9 is helpful, but it only shows 4 x MP fragments pots being sampled, and 5 x MP fibres, can you please check?

Line 374: how did you harvest the wheat? By hand with a tractor? Did you leave the stubble?

Line 376: ah I see that initial measurements were at n = 15 (wheat), and some of those were done at n = 5 (due to destructive sampling) leaving the subsequent experimental work at n = 10 (barley), but not the controls which were left at n = 15. Is that correct and if so, can you clarify that?

What happened with the remaining cores while waiting to be inserted back into the field after the wheat harvest? Can you explain storage conditions?

There was no addition of any fertiliser in between crop rotations - can you specify?

Line 372: Figure S10 is really helpful! It would also have been good to include some universal item to show scale. It does not show crop "progression" though, there is only an overview of the field showing that the wheat was there.

Line 388: can you specify how many soil columns? N = 5 for wheat and n = 10 for barley and n = 15 for control?

Line 391: with a high risk of cross contamination of indium, can you clarify what you did to prevent this (I would imagine dust and adsorption to equipment)?

Line 394: Quickly check if I still get it: the cores were 9 cm tall, so with the error margin of 0-1 cm and layers of 2 cm, there were 8/2=4 layers to consider of 2 cm and the deepest layer was there at 1 cm or not at all? Wheat had n = 5 for the treatments, so there were 10 pots in total, which were sliced into 4 or 5 layers, so 40 or 50 layers in total. Each layer had a technical replicate of n = 3, so there were 120 or 150 samples for ICP-MS. For barley this was double, because there were 10 pots left for the treatments, so 260 – 300 samples digested for ICP-MS. Is this correct? If not, then can you please make sure this is crystal clear?

What was done with the technical replicates – were they averaged?

Line 410: we would really like to know the "known" In-to-plastic ratio, because we don't know it. Particularly relevant would be how you made that standard curve, ensured accuracy and groundtruthed it. I know that relative errors are acceptable comparing *across* treatments (effect sizes) if these errors are kept consistent and everything else is the same, but the aim here is to reflect back on actual MPs in the soil layers. That would mean that we really would like to know the In-to-plastic ratio of the spiked soil too. Can you address this please?

Line 431: Great to see the check for leaching. Assuming that all of the added plastics remained in the mesh bags during the two years, would it have been useful to see if In was detectable in the soil directly under that bag?

Line 432 - 434: Are those values (in wt%) within the credible detection limit and error range of the equipment? Could do a quick and easy paired t-test, or similar test if the data are not parametric, to show that there was or was no significant decrease.

Line 439: here and elsewhere, can you specify the level of replication you employed for each measurement?

Line 498: which version of R was loaded in RStudio for the analysis? Could you also include references to provide credit to the developers where it is due?

Line 501: for objective 2) analysis over time, how did you deal with the non-independence of the samples and how did you deal with the unequal replication? What do you mean with "grouping the different treatments" please explain.

The one-way ANOVA really only works when you have means which are independent. It is not clear what the independent units are given the many steps and subsamples. You also potentially have $n = 5$, $n = 10$ and $n = 15$ so that will affect the degrees of freedom to calculate relevant probabilities from the test statistic. I thought the experimental design was nested (see line 371) at the biological (field) level, so how was the nesting taken into account in the stats?

Welch's tests use estimated degrees of freedom instead of the actual ones, can you explain how that works where you already have (I think!) very different degrees of freedom to start with (particularly when comparing across time, but also barley which appears to be $n = 15$ for control and $n = 10$ for treatments)?

Line 508: did you use a package for the Dunn test? I am not aware if native R has that test. If you use any packages, please cite the authors to give credit where it is due.

Line 509-511: I am afraid that p-values on their own are very uninformative. I also think that we should step away from the arbitrary reporting of p-values without any supporting information of how they are calculated. The minimum information for an ANOVA would be the F statistic, associated within (treatment) and between (residuals) group degrees of freedom. The treatment df for your design should be 2 in a one-way ANOVA, but the residuals are very dependent on how many replicates you have in total - and that is not made clear in the methods. Without these reported clearly, readers cannot double check the significance and the relevance of the tests, and have to take your word for it but it lacks transparency. Given that you also may have to consider the nestedness of your design in the field, you may need to calculate the F statistic based on the MSS and degrees of freedom of the random nesting factor usually a blocking factor in the traditional agronomy literature, but it is not clear that this was considered.

References

[21] needs to state author name(s).

Communications Earth & Environment is committed to improving transparency in authorship. As part of our efforts in this direction, we are now requesting that all authors identified as 'corresponding author' create and link their Open Researcher and Contributor Identifier (ORCID) with their account on the Manuscript Tracking System prior to acceptance. ORCID helps the scientific community achieve unambiguous attribution of all scholarly contributions. You can create and link your ORCID from the home page of the Manuscript Tracking System by clicking on 'Modify my Springer Nature account' and following the instructions in the link below. Please also inform all co-authors that they can add their ORCIDs to their accounts and that they must do so prior to acceptance.

Version 1:

Decision Letter:

Dear Professor Mitrano,

Your manuscript titled "A Two-Year Incubation Study: Investigating the Vertical Transport of Microplastics in Soil and Their Impact on Soil Pore Development" has now been seen by our reviewers, whose comments appear below. In light of their advice we are delighted to say that we are happy, in principle, to publish a suitably revised version in Communications Earth & Environment.

We therefore invite you to revise your paper one last time to address the remaining concerns of our reviewers. At the same time we ask that you edit your manuscript to comply with our format requirements and to maximise the accessibility and therefore the impact of your work.

EDITORIAL REQUESTS:

****Please take care to match our formatting and policy requirements. We will check revised manuscript and return manuscripts that do not comply. Such requests will lead to delays. ****

SUBMISSION INFORMATION:

OPEN ACCESS:

Communications Earth & Environment is a fully open access journal. Articles are made freely accessible on publication. For further information about article processing charges, open access funding, and advice and support from Nature Research, please visit <https://www.nature.com/commsenv/open-access>

Link Redacted

Best regards,

Somaparna Ghosh, PhD
Associate Editor,
Communications Earth & Environment

REVIEWERS' COMMENTS:

Reviewer #2 (Remarks to the Author):

The author nicely addressed most of my comments, well done. There is one comment open where I kindly disagree, I suggest a minor revision:

Line 51: The point about synthetic origin is unclear and needs precision. What are these "important questions"? Why is the synthetic nature of plastics the main issue? Would biobased materials present less of a problem? Please clarify these points, as the current phrasing is vague.

Rebuttal:

The synthetic nature of plastic prevents them from biodegradation. Unlike natural materials, MPs are not biodegradable, which means they can accumulate in soil and affect ecosystems over time. Biobased materials may improve biodegradability to some extent, although not all degrade as intended.

We rephrased the text (line 55-56) to convey the message more clearly and avoid confusion about synthetic origin: The resistance of MPs to biodegradation contributes to their persistence in the environment and may trigger unforeseen ecological changes.

I still do not agree with this suggestion:

Whether a plastic is biodegradable or not is solely a function of its molecular backbone and whether the bonds can be enzymatically broken (not if something is "synthetic"). Biodegradability has nothing to do with whether the source is based on renewable feedstock or fossil-based. For example, plastics based on renewable feedstock, such as sugarcane-derived polyethylene, are not biodegradable, while fossil-based PBAT, one of the major biodegradable polymers, is produced from fossil sources. Regardless, all of these plastics are produced synthetically.

This requires clarification by the authors: Plastics can be derived from fossil sources or renewable feedstocks, and they can either be biodegradable or non-biodegradable. While some authors argue that plastics based on renewable feedstock have a smaller ecological footprint, a) producing over 500 million tonnes of plastic from natural feedstocks would cause enormous harm to ecosystems, and b) the primary emissions stem from polymer production, not from the feedstock itself.

These aspects need clarification in the manuscript. The statement, "the synthetic nature of plastic prevents them from biodegradation, unlike natural materials," is, in my opinion, still incorrect.

First, I suggest you clarify the definitions of "based on renewable feedstock" and "biodegradable" in the paper. Second, I recommend including a sentence such as:

"Conventional non-biodegradable microplastics are persistent in the environment and may trigger unforeseen ecological changes."

Reviewer #3 (Remarks to the Author):

I would like to thank you for addressing all of my (and the other reviewers') comments on the original manuscript and associated content in such detail and in a professional manner, it is much appreciated.

I agree with your responses to my points, and thank you for clarifying aspects which were not clear for me.

Review of Manuscript:

A Two-Year Incubation Study: Investigating the Vertical Transport of Microplastics in Soil and Their Impact on Soil Pore Development

Corresponding author: denise.mitrano@usys.ethz.ch

We thank the reviewers and editor for their suggestions to improve the clarity of our manuscript. Below we have answered each comment in blue, where specific additions to the manuscript text are also included into the response to the reviewers, where appropriate, in italic font. In the response to reviewers' letter, we have also included references to line numbers where these additions can be found, where line numbers refer to the new, clean version of the manuscript. Along with this response to reviewers document, we have included a clean version of the manuscript, a tracked changes version of the manuscript and an updated supplementary information file.

Reviewer: 1

Comments:

This is a thoroughly conducted study on a very important topic, described in an excellently written manuscript. The authors use highlight sophisticated advanced X-ray CT tools to provide an in-depth assessment of micro-plastic addition effects on soil structure. The study is of great interest and importance for better understanding of the impacts of plastic pollution on the environment. Thus, I strongly recommend publication of the manuscript.

We thank the reviewer for their overall generally positive comments and for taking the time to suggest further changes to improve the clarity and impact of our work. Below we have provided answers to the reviewers' concerns.

Suggestions for improvement:

1. One major suggestion for improvement that I have is related to statistical analyses conducted in the study. The experiment has a very solid design with plenty of replications. It appears to be a 2-way factorial with factors: type of MP addition and the year of the observations. Plus, for some of the studied variables, e.g. transport related variable, the soil depth appears to be the third studied factor. Given all that, I am not sure I follow the reasons for the type of analysis (one-way ANOVAs) conducted here and whether it is really appropriate for this study. The effects of all these studied factors are either directly or indirectly mentioned and reported in the Results and involved in the Discussion, yet, the conducted one-way ANOVAs really cannot provide adequate statistical support for those. Why not do a proper 2-way factorial split-plot (or, better, repeated measures) analysis to address significance of the treatment and year factors and their interactions? Or 3-way split-split-plot analysis with depths as a repeated measures? That would greatly strengthen the study! One other related question I have is the choice of the non-parametric tools for the analysis – from the graphs (e.g., Fig.4) it looks like often it might be just one or two extreme outliers that drastically increase variability and lead to the lack of statistical significance. I would recommend exploring either excluding these outliers or transforming the data prior to the analyses. I believe that might generate results with a greater statistical power.

Thank you for your valuable feedback regarding the statistical analyses. We have reanalyzed the data using linear mixed-effects models (LMMs) to better reflect the factorial design of the study. The models included the effects of Replicate, ROI, and Year for morphological measures, and Replicate, Layer, and Year for vertical transport of MPs, with random effects to account for block variability. Post-hoc pairwise comparisons were conducted using the emmeans package with Tukey's HSD adjustment for multiple comparisons. Outliers were identified using the interquartile range (IQR) method and excluded if they fell outside 1.5 times the IQR, provided they were confirmed as true outliers and not biologically meaningful (i.e., plausible or valid biological data). Data transformations (e.g., log transformation) were applied where necessary to address skewness (only for vertical transport data). We updated the method section to reflect these changes regarding statistical analysis. The revised statistical results are now included in section S11 (SI), with tables summarizing the analyses and updated significance levels. We believe these changes provide a more robust statistical framework and address your concerns.

2. It would strengthen the relevance and importance of study to provide a justification of how the plastic materials created here are relevant for typical plastic pollutants found in the soil.

Thank you for the suggestion. While we acknowledge that polyethylene, primarily from agricultural mulching films, is the most prevalent polymer found in soils, we still believe that PET fragments and fibers serve as an excellent model for studying the environmental impacts of agricultural plastics. PET MPs, commonly found in textiles and packaging, are frequently detected in terrestrial environments and represent a relevant and increasingly recognized form of plastic pollution. By focusing on PET, we aim to explore its behavior and impact in soils, providing insights that are applicable to a wider range of plastic pollutants. We feel this choice offers a balanced approach, aligning with the scope of our research while highlighting the significance of various plastic types in the environment.

We have added the following lines (129-130):

Our study focused on PET fragments and fibers, common forms of MPs as model materials to explore MPs pollution in soils.

And the following lines (400-404) to the Methode section:

While polyethylene is the most common polymer in soils, mainly from agricultural mulching films, PET is still a good choice as a model for studying plastic pollution. PET fragments and fibers, often from textiles and packaging, are increasingly found in terrestrial environments.^{1,2}

3. The separation into the three regions of interest (ROI1, 2, and 3) makes perfect sense, however, having these regions described only in the Methods makes it impossible to interpret the Results (which come prior to the Methods). I would recommend adding a brief description of these three regions early in the Results. Also, it might be helpful for the reader is using more meaningful names for them - e.g., "entire core" or such for ROI1, top of the core... instead of just umbered ROIs

This is a very valid point. We updated the Results section by adding a short description of the layers and ROIs which were used for vertical transport analysis and also for X-ray analysis.

We have added the following lines (145-155) to the Results section:

For soil morphology analysis of the columns, three different regions of interest (ROIs) within the soil column were examined. ROI 1 encompassed nearly the entire soil column, measuring 6.5 cm in height, and was further divided into two subregions: ROI 2, which represented a small part (1 cm height) of the top of the core where MPs were initially added at time zero, and ROI 3, which consisted of the lower 4 cm of the core where we anticipated less MPs accumulation. Separately, vertical transport of MPs in the soil columns was assessed across five layers, which were physically sectioned and further analyzed

for MPs content. The uppermost four layers were each 2 cm thick, and the lowest layer varied between 0-1 cm, depending on the compaction of the soil over time (Figure 1). Further details regarding the ROI for microtomography and the sectioning of the soil columns can be found in the methods section. Exact p-values, point estimates, standard errors and degrees of freedom of statistical analysis of all morphology measures are summarized in section S11 (SI).

Minor questions/comments: For soil morphology analysis of the columns, three different regions of interest (ROIs) within the soil column were examined. ROI 1 encompassed nearly the entire soil column, measuring 6.5 cm in height, and was further divided into two subregions: ROI 2, which represented a small part (1 cm height) of the top of the core where MPs were initially added at time zero, and ROI 3, which consisted of the lower 4 cm of the core where we anticipated less MPs accumulation. Separately, vertical transport of MPs in the soil columns was assessed across five layers, which were physically sectioned and further analyzed for MPs content. The upper four layers were each 2 cm thick, and the lowest layer varied between 0-1 cm, depending on the compaction of the soil over time (Figure 1). Further details regarding the ROI for microtomography and the sectioning of the soil columns can be found in the methods section. Exact p-values, point estimates, standard errors and degrees of freedom of statistical analysis of all morphology measures are summarized in section S11 (SI)

1. L.148 – What ROI is shown on Fig. 3? Seems like a discrepancy between the text and figure caption.

Figure 3 indeed shows ROI 1. We updated the text (line 177-179) to make it clear that we speak about ROI 1 as follows:

This trend was likewise evident in the three-dimensional representation of temporal biopore evolution in the whole ROI 1 of a control column (Fig. 3, panels: a-c).

2. Fig. 2 – Please specify the lower boundary of macropore size. It is not zero, right?

We defined the lower limit for macropore diameters, to be at 0.3 mm, following Jarvis (2007). This threshold corresponds fairly well to our image resolution. In the methods section it is already stated that we have an approximate image resolution of 300 μm . Therefore, we believe that the manuscript accurately reflects that the lower limit of the macropore size is not equal to zero.

3. 1.173 Fig. 3 - Would not "Visible on the images" be clearer? Also, my understanding is that time zero scans are for initially packed cores – how come they already have so many biopores. Are these definitely biopores?

The soil columns were filled with soil on the 11th October, put in the field to let drain over a 5-day period and then we stored the samples in a dark cold room at 4° C until X-ray scans were done on the 28th of October. Time point zero represents the first X-ray scan that took place on the 28th of October. Indeed, there were biopores within the soil aggregates used to pack the soil columns at time 0 because of this slight delay in analysis. This is because new biopores were dug in the 5 days of pre-incubation in the field. Please also note that while the biopore detection algorithm provides a reasonable estimation of the biopore volume fraction imaged, and it is by no means perfect and occasionally produces false positive detections.

4. 1.204 – Is this correct that here “transport” is actually the concentration of MP found at different depths? If so, would be better to say so.

We have adapted line 243 accordingly with the phrasing: “, *with slightly increased MPs concentration* ...”

5. Fig. 5 – Marking and mentioning statistically significance in terms of treatments, years, and depths would be beneficial here.

We have updated Figure 5 to include the results from the statistical analysis overall, highlighting the effects of layer, replicate, and year on the dependent variable. Additionally, we have directly annotated the pairwise comparisons for replicates on the plot, marking statistical significance with asterisks.

6. 1.487. Please provide formal definitions and determination methods for microporosity and specific surface area and 1.494 for pore size distribution.

Definitions have been added for macroporosity, specific surface area of macropores, and of pore size distribution, including determination methods directly in the manuscript.

New text line 586-588: *Macroporosity is defined as the ratio of the macropore volume (void spaces) to the total bulk volume of the soil and the specific surface area of macropores is the total surface area of the macropores per unit volume of soil.*

Adapted line 590-593: *Additionally, we computed the pore size distribution using the maximum inscribed sphere method ('Local Thickness') as implemented in SoilJ and the distance of a soil matrix voxel to the nearest top-surface-connected macropore, the latter providing a proxy for soil aeration.*

7. 1.503-504 What is the purpose of determining "significant differences between replicates"? Or is it just incorrect description?

We apologize for the confusion. This aspect was indeed not described well and was meant to be about significant differences between group means across different treatment or condition groups. The whole statistical analysis has been changed upon request from all reviewers, therefore the whole statistical analysis section has been adapted accordingly in the manuscript and this specific comment is no longer applicable.

Reviewer: 2

Comments:

The transport of plastics in soil, particularly in agricultural systems, is an important issue. While much research has been conducted over the past decades, the paper by Schefer et al. makes a useful contribution by employing labeled plastics to enhance our understanding. I recommend the manuscript for publication, but only after major revisions. While the results are largely predictable, the use of labeled plastics represents a valuable methodological advance.

We thank the reviewer for their overall positive feedback. We hope that the answers to their queries, along with changes to the manuscript, help to improve the understanding and quality of our work.

Abstract: The assertion that plastics are primarily used unintentionally in agriculture is inaccurate. Please address and expand upon this misconception in detail within the revised paper. The main use is intentional.

We have now changed the text in the abstract to the following (l. 12-14):

Plastics are widely and intentionally used in agriculture to improve productivity and resource efficiency. However, as these plastics fragment over time, microplastics (MPs) are unintentionally released into the soil, raising concerns regarding their long-term implications for hazards, soil structure and fertility.

Line 19: Remove the word "significant" as it reflects incorrect statistical interpretation of p-values.

Removed “significant” in the respective line.

Line 21: Clarify that microplastic transport was very limited, with observations restricted to the top few centimeters, and only 1% reaching deeper layers. This crucial finding from the results must be included in the abstract.

Adapted the text to include the main finding by adding following text (l. 22-24):

MP transport was minimal, with nearly all MPs remaining in the top few centimeters of soil and only approximately 1% reaching below 8 cm after two years, regardless of MP morphology.

Line 25: The statement overstates the paper’s relevance. The manuscript focuses on plastic transport, soil structure, and macropore evolution, but these are just a few aspects of plastics' broader impact on soil. You do not, for example, address the release of additives or effects on plant productivity and soil microbiomes. Please moderate the claim to reflect the paper’s specific focus. Your conclusion handles this well—consider adopting a similar tone here.

We lowered the tone of the statement in the abstract by the following text (l. 25-29):

This experimental design, mirroring natural soil conditions and incorporating soil biota, allows for the determination of realistic MPs transport rates and offers insights into how plastics impact soil structure, a key factor in understanding the broader impacts of MPs on soil ecosystems. Overall, the findings suggest that MPs have a negligible influence on soil macropore architecture in the short term.

Line 34: Consider removing normative jargon such as "mindfulness." The term is overused and detracts from the scientific tone.

Adapted the sentence to the following (l. 38-39):

However, it is important to recognize that the immediate advantages of plastic utilization may come at the expense of long-term sustainability goals.

Line 43: Avoid terms like "pivotal." Soil fertility has many contributing factors, and plastics are just one aspect. The paper tends to overemphasize its findings—please moderate these claims and use a more neutral, matter-of-fact tone.

Thank you for pointing it out! We changed it to “important” in line 47.

Line 51: The point about synthetic origin is unclear and needs precision. What are these "important questions"? Why is the synthetic nature of plastics the main issue? Would biobased materials present less of a problem? Please clarify these points, as the current phrasing is vague.

The synthetic nature of plastic prevents them from biodegradation. Unlike natural materials, MPs are not biodegradable, which means they can accumulate in soil and affect ecosystems over time. Biobased materials may improve biodegradability to some extent, although not all degrade as intended.

We rephrased the text (line 55-56) to convey the message more clearly and avoid confusion about synthetic origin:

The resistance of MPs to biodegradation contributes to their persistence in the environment and may trigger unforeseen ecological changes.

Line 84: There is a substantial body of research on particle transport in soil. While you cite a few

relevant papers, the existing literature on older colloidal transport (e.g., Kretschmar, Ryan, Elimelech, and others) is more comprehensive and should be referenced. The claim that nanoplastics would be more mobile is incorrect; according to filtration theory, particles around 1 μm in size are most mobile. Smaller particles, like nanoplastics, are likely less mobile. Please revise this statement.

We revised the paragraph accordingly and included some of the references the reviewer suggested, as per below (l. 96-104):

Literature on colloid transport in porous media has shown that particles in the size range of 1 μm tend to be more mobile than smaller or larger ones. Larger particles ($>1 \mu\text{m}$) are prone to straining and physicochemical interactions (van der Waals forces, electrostatic interactions, and hydrophobic effects) that hinder their movement, while smaller particles, such as nanoparticles, may encounter increased retention due to electrostatic and surface interactions with the porous matrix.³⁻⁵ Nano-plastic may however also adsorb to soil colloids that are more mobile and may be transported in this fashion. Additionally, for MPs, particles with complex, nonspherical shapes exhibited higher adhesion to porous media, reducing their mobility compared to more spherical or simple shapes.^{6,7}

Line 98: The term "unravel" overstates what can be achieved. You are contributing to the understanding of certain aspects, not resolving all complexities. Please temper this statement to be more realistic and precise.

We tempered the statement as follows (l. 116-118):

In this current study, we aimed to explore the effects of MPs on soil pore structure by investigating the transport of MPs fragments and fibers in soil under field conditions and assessed its impact on the formation of soil macropore network structures over two years (Fig. 1).

Line 163: I disagree with the statistical analysis. Please see my comment on Line 497 for further details.

Thank you for your feedback regarding the statistical analysis. As noted in our response to the comment on Line 497, we have revised our approach to address the limitations of relying solely on p-values. The updated analysis now includes effect sizes and confidence intervals alongside p-values, which are reported in detail in Section S11. These changes reflect the recommendations from Amrhein et al. (2019) and the ASA guidelines (as indicated by the reviewer in a comment below) for a more comprehensive interpretation of statistical results. While we have retained significance markers in the figures for clarity, we have ensured full transparency by providing the exact p-values, effect sizes, and confidence intervals in the supplementary materials. We hope this revision addresses your concerns and enhances the clarity of our statistical reporting.

Line 201 and following: The finding that microplastics are relatively immobile in soil is consistent with theoretical predictions from prior research. I appreciate the innovative approach of using labeled microplastics to confirm this.

Thank you for your comment. We appreciate your recognition of our approach in using labeled microplastics to confirm their immobility in soil.

Line 497: I strongly disagree with the use of p-values to determine statistical significance. The concept of statistical significance based on p-values has been extensively critiqued. Please refer to Amrhein et al. (Nature, 2019) and the ASA's guidelines on this issue. It is incorrect to assert that a p-value of 0.05 defines significance. I recommend fully revising the statistical analysis in accordance with current guidelines. Compare: <https://www.nature.com/articles/d41586-019-00857-9> and <https://reproductive-health-journal.biomedcentral.com/articles/10.1186/s12978-021-01131-w>

We fully agree with the reviewer's concern that relying solely on p-values to determine statistical significance is insufficient and potentially misleading. As noted in the literature, including Amrhein et al. (2019) and the ASA's guidelines, the interpretation of p-values should be placed in context,

alongside considerations of effect size and confidence intervals. In response to this, we have revised our statistical analysis accordingly. The updated analysis described in the method section “5.9. Statistical Analysis”, includes detailed reporting of exact p-values, effect sizes, and confidence intervals, all of which are transparently presented in tables in the supplementary information.

While we acknowledge the limitations of using p-values in isolation, we believe that for clarity, the use of asterisks to indicate significance can still be helpful for readers when interpreting the plots. Therefore, we have retained significance levels in the figures, but have ensured that the exact p-values, effect sizes, and confidence intervals are available in the supplementary materials for full transparency and comprehensive interpretation.

Reviewer: 3

Synopsis:

This manuscript reports the findings from a field study, where cores containing soil spiked with PET microplastics which were doped with indium were exposed to an agricultural setting growing wheat and barley. The study employed state of the art techniques to track where microplastics ended up in the soil profile of the cores, but also explored the soil porosity and biological aspects thereof at three time points. The study reports that there were minimal effects overall when comparing the treatments to the controls, but over time the pore structure was different in the controls. Furthermore, levels of indium as a proxy for microplastics were found in deeper soil layers, particularly over time, but perhaps not as much as the authors expected which suggests that the majority remained in the initially spiked top layer. The authors discuss the findings in light of biological and physical processes, suggesting that root growth but also possible burrowing by animals may explain this observation. The authors conclude that sustainable alternatives [to conventional plastic types] should be explored to reduce the effects of microplastics in soils.

I really enjoyed reading this manuscript, it is presented at a very high quality, which would be expected. The amount of effort which has gone into this piece of research is impressive! The research questions are interesting and will be of great interest to researchers in the discipline, particularly how the authors have employed ways to explore soil structure and pores in situ (which is not that new in itself, but a great way to do see what's inside). The indium tracer technique used appears to have been employed successfully by the research group previously and the authors report that there are appropriate quality controls in place. There are however some points which are concerning me; they are not necessarily related to the methods and associated analytical aspects, but more so on the wider implications, the experimental design and implementation, statistical analysis and the presentation thereof, and the quality of the discussion. I have laid out the summarising points in the main comments below, and I took the liberty to provide more in-depth comments and suggestions to provide more details to support the main comments, but also some points which could be clarified more in my opinion.

Major comments:

1. The statistical analysis needs much more transparency to show the readers what was done exactly. The methods section suggests that the number of replication changed over time, which is acceptable, but you will have to show how that affected the degrees of freedom (for the ANOVA) and thus the calculated p-values upon which conclusions are based. P-values on their own are rather uninformative and murky at best. Furthermore, there are subsamples as well to measure response variables, probably as technical replicates, but it is not clear what you did with those to assess the measurements at the biological level. For example, with the layers included, readers could interpret the methods that up to 300 ICP-MS measurements were done, but this is quite unclear. Throughout the methods it would be imperative that all replication is clearly explained.

Thank you for your feedback. We have improved transparency in our statistical analysis in the text in a number of sections, as outlined in the detailed point-by-point responses to all reviewers. Additionally, we now included point estimates and confidence intervals to provide a more comprehensive understanding of the results. These updates are presented in the supplementary section S11. We have also revised the methods section to clarify the use of technical and biological replicates, ensuring the replication structure is fully explained.

2. The methods also state that a nested field design was employed, supported by figure S9 (please check this figure for the removed cores!). These are standard agronomic field tests, nothing wrong with them, but they are nested to account for random spatial variation. Some call them “blocks”, yet there is no clear consideration given in the statistical analysis for this nestedness. It will affect how the degrees of freedom and which sums of squares are used for the parametric test. This also brings me to the point of non independence of time points, the cores, while removed in year one, essentially remain the same (bar from removing $n = 5$ from the treatments), and are put back into the soil. Whatever was there in $t = 0$ probably affects $t = 1$, and that is carried over to $t = 2$. The statistical analysis ideally should take this into account, but that is not made clear and therefore I suspect it was not considered. It will probably require consideration and could lead to a change in the significance of some results.

We accounted for the nested field design by using "Blocks", which group measurements from the same nest in our statistical analysis by a linear mixed model. Each block contains three treatments (control, fragment, fiber) and measurements were taken across three time points (year 0, year 1 and year 2). The repeated measurements within each block were captured by including “1|Block” in our statistical model, which accounts for the non-independence of time points. This approach ensured the nested structure was properly considered in the analysis. We updated the statistical analysis in the methods section and also included transparent output from the statistical analysis in supplementary information.

3. The cores included three very big plants (monocots) which of course would have a major influence in producing pores due to their rooting. While this does not invalidate the findings, the interpretation thereof needs care, because there study design does not exclusively test the effects of microplastics on pore formation, instead, as a whole, it tests any direct and indirect effects (on the plants). I think that reporting non-significant results are just as important as significant ones, but they still need to be contextualised just as carefully. If the aim was to simulate an agricultural setting, with all relevant aspects for as much feasible, then that has been achieved – the unravelling of interactions, however, particularly mechanistic ones, are not that clear however. For that a set of unplanted cores would have been needed, which of course would double the amount of work.

Thank you for this insightful feedback. Our goal was indeed to simulate an agricultural setting as realistically as possible, incorporating the complex interactions found in such environments, including those influenced by large monocot plants. We recognize that these plants contribute considerably to pore formation through rooting activity, which could impact the interpretation of microplastic effects on soil pore structure. However, mechanistic understanding of processes through which MP impact soil structure, or impacts on plant growth or rooting, was not our goal: we used this design to simultaneously assess if a) the presence of MP had any effect on the pore structure evolution under realistic conditions and b) quantify the transport of MPs under realistic conditions. Our study, therefore, captures both direct and indirect effects of microplastics on the soil ecosystem as a whole, reflecting the complexity of real agricultural systems. We agree that including unplanted cores or cores from which the soil macrofauna was excluded or both could have provided a clearer mechanistic understanding by isolating the effects of microplastics on pore formation (as well as, incidentally, teasing out how important bioturbation from the monocots and/or macrofauna was for transport by other factors). While such experimental designs are very interesting options to advance this field of research, since our resources in time, money and workforce were limited, we were unable to complete additional variables as suggested by the reviewer.

We further appreciate the point about non-significant results. We've carefully reported both significant and non-significant findings in the context of this complex system, ensuring they are appropriately contextualized to provide a balanced interpretation. This was particularly important to show there were no observable impacts from microplastics on soil structure when there are multiple other factors in play: a result which may not have been seen if only working in a small, short-term, laboratory study – or without the presence of soil biota.

4. Were there any soil animals present in the cores? I know it was not part of the aim, but could be interesting to explain some findings due to homogenisation (worms are particularly good at that).

In fact, our study design provided access for earthworms and other soil macrofauna to enter and exit the soil column. Moreover, the experimental site was chosen due to its reported large abundance of earthworms. And yes, we observed worms in the soil cores during the sectioning process, as they were identifiable by eye and visible in the X-ray images acquired. While our study did not aim to differentiate worm burrow systems from other types of biopores, the diameter and morphology of many biopores point towards earthworms as their creators. The presence of these organisms has indeed very likely contributed to a considerable degree to soil bioturbation and, thus, homogenization. Given the low mobility of the MPs investigated here by advective transport³⁻⁵, the observed displacement of MPs in the soil is best explained by transport with earthworms and/or movement through plant root growth in our study. Additionally, we found evidence of other soil-dwelling organisms, such as beetles, were also in the soil cores. However, further investigation into the specific impacts of these organisms was beyond the scope of our study.

5. Technically, the effects of microplastic fibres and fragments WITH indium was assessed, not microplastics without the added metal, how little it may be. I completely understand that the idea was to track the plastics using indium as a proxy for the plastics assuming no leaching occurred from the plastics (which you have controlled for, that is great!), but this additive could be considered as an extra aspect in the design. The plastics are obviously model particles, because according to the methods they have been processed after purchase and modified to fit the purpose of the study, therefore it should be noted that these particular model plastics are not necessarily reflective of what is typically found in an agricultural soil. This aspect does not invalidate the results, but it does require acknowledging the translatability of such models to a wider context even if it was carried out in a more natural setting than a controlled lab study otherwise. It could have been done with plastics which were not metal-doped, but of course then you have the issue with not being able to track them (but can still assess the pore formation). I suppose this would be a “procedural control” for the manipulation of the experimental units, which are the plastics.

We thank the reviewer for their pragmatism in recognizing the materials we made were suitable for purpose, and can appreciate the concerns they may have regarding the translatability of our model plastics compared to “regular” (micro)plastics. Indeed, this is the point which is most commonly raised when we have used these materials in numerous studies before.

With regards to the material itself, as the reviewer recognized, the actual metal content is very low (0.2 w/w) so the materials' density is negligibly impacted, and it's important to mention that the metal is inside the MPs and not on the surface. So, the surface chemistry of the material is the same as any other MPs which would be produced via cryomilling, which is one of the most common ways to make MPs for research purposes. Furthermore, we have tested (in this study, and in others) that there is no leaching of metal over time.

Despite our intentional incorporation of a known amount of metal which was quantified, these results gained from these materials are transferable to other (micro)plastics because the term plastic itself describes a very heterogeneous group of materials. Indeed, the incorporation of metal into plastics is a

common industry practice (e.g., as a catalyst, colorant, or other purposes). While we have not measured the metal content of (micro)plastics in agricultural soils, from other (as yet unpublished) work, we have assessed the metal distribution across a range of plastics collected from the Great Pacific Garbage Patch in the Pacific Ocean and found that there were a range of native metals included into a variety of different products. Therefore, the materials we use can be considered as a “normal” plastic.

6. I understand that the crop was harvested too, and used to assess yield (see methods), but the results are not presented here. I do understand that this was not part of this particular project’s questions, but in light of the claims related to extrapolation to field settings, particularly with the aim in mind of informing “decision makers”, this finding would be of interest.

We added the following section in the material and methods part to briefly address crop yield results, as we did not wish to place primary focus on this aspect in the results and discussion portion of the study (l. 520-529):

5.8 Crop yield analysis of wheat and barley

The hand-picked wheat and barley crops from the columns were dried at room temperature for a minimum of four weeks. Following drying, the grains were separated from the remaining plant material. Total biomass, the grain weight, and the weight of the residual plant material (e.g., straw) was measured. The harvest index was calculated as the ratio of grain weight to total biomass. The results suggest that the addition of MPs had very little to no effect on the crop yields of wheat and barley overall. Although the presence of MPs fibers likely results in a slight reduction in total barley biomass and grain weight compared to the control (estimate total biomass = -10.2 g, $p = 0.051$ and estimate grain weight = -4.28 g, $p = 0.0625$). All results from crop yield analysis are presented in Figure S12 in the supplementary section and the statistical analysis in section S11 (SI).

7. Even though the experiment encompassed an impressive two years AND was done in the field, the experimental units were still rather confined, with the cores being 10 cm diameter and 9 cm deep – particularly when realising that each core supported three plants and their rooting system for two years. I think this aspect could be considered when discussing the findings as well.

We can appreciate the reviewer’s comment. The number of plants per sample was chosen following the advice of Jochen Mayer (Dr. agr., dipl. ing. sc. agr. at Agroscope, Zürich). The plant density outside the columns was similar to that within the column. The aluminum rings chosen for this work had openings in the walls to allow roots to grow outside of the sample in a realistic fashion. In our study, we used X-ray imaging. The image resolution of modern industrial X-ray scanners scales with the diameter of the investigated object. Moreover, there is a restriction on the maximum diameter of a soil sample that can be imaged with X-rays, since it needs to be thin enough to allow a sufficient number of X-ray photons to traverse the sample. The size of the columns chosen in our study represents a compromise between including a soil volume as large as possible for the sake of representativeness and restricting the sample diameter to enable scanning and obtaining a resolution with which the soil macropore network could be mapped in high detail. In addition to the fact that it would have necessitated us to produce a lot more MPs in order to include in the system, and they may be further diluted in the soil layers which would increase the limit of detection. However, as only 1% of the MPs reached the bottom of the column after 2 years (and, we still had a high recovery (i.e., mass balance) of the MPs which were initially put into the column, we are very confident that this was very appropriate column size for our experiments in terms of assessing the transport and soil macrostructure.

Minor comments and suggestions for change:

Title:

Informative, but please check that it clearly conveys the message: with “its” do you refer to vertical transport (singular noun), or microplastics (plural noun), or the impact of “investigating” as a whole? I know this is rather pedantic, but I also think it is important to avoid confusion with the first thing someone will read.

Thank you for pointing this out! We slightly adapted the title by adding “their”, to more accurately convey the intended message:

A Two-Year Incubation Study: Investigating the Vertical Transport of Microplastics in Soil and their Impact on Soil Pore Development

Keywords:

I wonder if “metal-doped plastics” is probably not a string of keywords someone will use when searching for this particular type of research. Just a thought.

We deleted the key-word “metal-doped plastics. And made keywords to: Pollution, Environment, Macroporosity, X-ray, Bioturbation, Field study

Abstract:

line 12: why only “unintentional”, could be any type of introduction, intentional or not?

As already similarly pointed out by reviewer 2, we changed to text in the abstract to show that plastics are intentionally used, but then unintentionally get released into soil as smaller fragments. See new text (l.12-14):

Plastics are widely and intentionally used in agriculture to improve productivity and resource efficiency. However, as these plastics fragment over time, microplastics (MPs) are unintentionally released into the soil, raising concerns regarding their long-term implications for hazards, soil structure and fertility.

Line 13: perhaps remove “fertility” given that this is not the focus of the study (fertility sensu stricto was not measured as part of the study)?

Although soil fertility was not specifically measured in this study, we included this statement to provide context on why studying microplastics in soil environments is important. In our opinion, mentioning potential concerns about long-term implications for soil fertility due to plastic pollution adds relevant background to the abstract and supports the study's broader significance.

line 21: were the pores in the controls different from the treatments, or were the significant changes over time (or indeed, both)? Can this be clarified in the abstract given what is stated in line 23?

The pores in the controls were only slightly different from the MPs treatments in ROI 2 in year 0 and year 1 as indicated in Figure 4 in the manuscript. Although the differences were statistically significant, the actual difference was very minor. Our rationale behind only showing the control samples for the overall trend of soil structural changes over time was that we wanted the same number of samples during all sampling points and to simplify the story.

We adapted the abstract by adding the following lines (l. 20-22):

Both control and MPs-containing sample showed changes in soil structure over time, with shifts towards larger macropores, increases in biopore volume fraction and critical pore diameter.

line 25: what is the take-home message (impact) from the study, other than this allows for

determinations of realistic transport rates on soil? I would appreciate a more impactful concluding statement.

We refined the last two sentences of the abstract to deliver a clearer take-home message, as follows (l. 25-29):

This experimental design, mirroring natural soil conditions and incorporating soil biota, allows for the determination of realistic MPs transport rates and offers insights into how plastics impacts soil structure, a key factor in understanding the broader impacts of MPs on soil ecosystems. Overall, the findings suggest that MPs have a negligible influence on soil macropore architecture and transport is limited in the short term.

Introduction:

Line 28-31: perhaps this sections about the UN-SDG is not relevant for the nature of the study, I appreciate it is *important*, but it may not be needed to introduce this highly focusses study. Perhaps jump straight into the issue of soil structure and how there is evidence of how microplastics may affect that?

We refined the start of the introduction by taking out the UN-SDG, but we still kept a similar structure, since we are convinced that it is important for the reader to know why plastics are used in agriculture. Please see the new lines (l. 34-39):

Amidst rapid industrialization and a growing global population, responsible resource management is increasingly emphasized, particularly in critical areas such as food and water. In agriculture, ensuring robust crop yields and efficient soil and water management are paramount. Plastics have emerged as a significant contributor, seamlessly integrating into modern agricultural practices to support these goals. However, it is important to recognize that the immediate advantages of plastic utilization may come at the expense of long-term sustainability goals.⁸

Line 43: Not a huge issue, but agronomists maybe disagree that fertility sensu stricto is not necessarily soil structure, nor vice versa. Of course, soil structure is important to make a soil more accessible to resources. I would be careful with this and the wide readership in mind of the journal.

We agree and have toned down the statement accordingly.

Line 48: are you sure that “evolution” would be a correct term (given the traditional connotation with the biological adaptation to biotic and abiotic parameters), or could we think of this as ‘development’?

Yes, we are convinced that “evolution” is the right term for it. As it was also successfully published before in the context of soil morphology changes. See for example: Koestel et al.⁹, Keller et al.¹⁰

Line 51: why is it the “synthetic origin” that raises these questions?

Thank you for pointing it out! Reviewer 2 had similar concerns. We adapted the text so that it conveys the correct message as seen below (l. 55-56) :

The resistance of MPs to biodegradation contributes to their persistence in the environment and may trigger unforeseen ecological changes.

Line 54: what are the “qualitative” effects these sources [21-24] demonstrated on “soil systems”, can you give an example? Why specifically “qualitative”?

You are totally right with your indication, and we believe that qualitative might not be the right term. Those previous studies have primarily assessed the effect of microplastics on soil pore structure via an

indirect approach by measuring soil hydraulic properties (such as tension, water infiltration, and saturation). While soil hydraulic properties are undeniably essential, comprehending and forecasting soil hydraulic behavior necessitates the incorporation of not only the direct assessment of hydraulic properties but also the underlying soil properties (e.g., soil texture, porosity) that exert an influence on them.

Therefore we changed the wording to “*indirect*” effects in the text.

Line 56: I see that you try to justify the novelty of a long-term field study here, but since these sources [21-24] much more has been done (there are a lot of scientific studies on this topic now), including field-based work which has appeared recently. This could be made much stronger by also emphasising newer studies to show a wider picture of what has been done in this regard. In fact, you cited [27] later on for this?

Thank you for your feedback. We have included a brief paragraph highlighting literature about more recently published field studies. Recent field studies have primarily focused on the accumulation of microplastics in agricultural soils, with only a few investigating their effects on soil properties. See the new added lines including their references (l. 60-70):

Long-term field studies (duration of 2 years and more) have focused mainly on the accumulation of MPs in agricultural soils, with only few examining their impact on soil properties. MPs were found to alter soil pH, moisture retention, and nutrient availability, notably increasing nitrogen levels while decreasing the C/N ratio. Furthermore, MPs disrupted the structure of soil microbial communities, promoting the growth of pathogenic microbes and reducing the diversity of beneficial microorganisms. These alterations in soil physical and biological properties were linked to reduced crop productivity and water use efficiency, with plastic residues also inhibiting root growth and water movement.¹¹⁻¹⁶ However, the effects of MPs on soil morphology over extended periods in natural field settings remains less explored. To better assess the impact of MPs on soil morphology, quantitative analyses over more extended exposure periods are essential, necessitating in situ studies under field conditions.

Line 60-61: “quantifying MPs transport through soil depth over time will help constrain risk assessments by better understanding temporal exposure.” is quite vague, at least I had to read it a few times to get a sense of what you actually mean: what are risk assessments and how are they conducted with the added knowledge? What is temporal exposure – is there any other type of exposure we do not understand that well? I think I know what you mean but for clarity, this could be reworded.

If we know where microplastics accumulates and how fast it goes downwards, we can better anticipate the concentration of microplastics at different soil depths. We improved the clarity of the sentence as follows (l. 72-74):

Furthermore, quantifying MPs transport through soil depth over time will help improve risk assessments by providing insights on the timescales MPs move through the soil profile and under which conditions.

Line 96: Has this technique been used before with similar particles? If so can you add another source to the establishment of the method, if not, then we will be curious to see the quality controls in your study (which should be there anyway, but even more relevant for method validation).

Yes, this technique has been successfully used in several studies, and a number of them have already been referenced in this present manuscript. One study focused on extraction methods for metal-doped microplastics from various model and standard soil matrices, another explored extraction from unsaturated porous media in sewage sludge applications, while a third study extracted metal-doped microplastics from activated sludge in a municipal wastewater treatment plant. The final study

investigated the extraction of metal-doped nanoplastics from soil. We added the required references in the text (l. 114).

Each ICP-MS measurement included the use of internal standards (Rhodium and Palladium at 1 µg/L). Additionally, a quality control run with 1 µg/L Indium was performed after every fortieth sample to ensure the accuracy and precision of the analysis. (Also mentioned in the method section 5.7)

Line 98: I don't think you have really explored the predicted or possible "interactions" that much to unravel – what exactly do you refer to with *interactions* for this study (that they somehow cause more pores? Bigger pores, smaller pores? That plastics go through pores become adhered to pore walls?)

Your concerns are entirely valid. Our study did not aim to identify specific interactions between microplastics and soil pore structure but rather focused on examining the effects of microplastics on soil pore characteristics. We are unable to determine the underlying processes driving these effects with the current analysis we performed. In retrospect, the term "interactions" may have been misleading; a more accurate description would be the "effects" of microplastics. Specifically, we investigated to what degree the presence of microplastics impacts the development of macropore structural features like pore sizes, pore quantity, and pore connectivity within the soil

We therefore changed the lines 116-118 to the following:

In this current study, we aimed to explore the effects of MPs on soil pore structure by investigating the transport of MPs fragments and fibers in soil under field conditions and assessed its impact on the formation of soil macropore network structures over two years (Fig. 1).

Line 99: can you explain how unravelling these expected complex interactions will lead to informed decision making? What will this information change for making any current decision? While I understand that impact is important, statements such as these will need to be supported and clarified. Almost everyone can say that their research will 'inform decision making', but not many follow through which type of decisions may be affected and by who will make the decision (farmers? "the public"? local government? UN?). Banning plastics altogether, because they have (complex) interactions with soil pores? Swapping for biodegradable plastics because... I know this may come across perhaps a bit harsh, but I think it is important to consider the implications of such statements!

Thank you very much for your insightful and constructive feedback. We now see the problem with this statement more clearly and agree that it lacked specificity and could be misleading without clear examples of how the findings would directly influence decision-making or who the intended decision-makers are. Your critique is entirely valid, and we appreciate you bringing this to our attention. Upon reflection, we have decided to remove this part of the statement from the manuscript, as we agree it doesn't adequately reflect the current state of our research. While understanding the impacts of microplastics on soil pore structure is an important scientific question, any claims about informing decisions should be based on clear evidence, which we are not yet able to provide.

The new start of the paragraph is adapted to the following text (lines 116-118):

In this current study, we aimed to explore the effects of MPs on soil pore structure by investigating the transport of MPs fragments and fibers in soil under field conditions and assessed its impact on the formation of soil macropore network structures over two years (Fig. 1).

Line 102: what is an "extended two-year timeframe" is that with "extended" more than two years? If so, can you be specific please? Could you just say "over two years"?

Indeed, we can simply say “over two year” instead of “extended two-year timeframe”. Our aim was to highlight the 2-year duration, which examines the effects over a longer duration than most previous research. We adapted the text accordingly.

Line 102: are elements always with a capital letter when spelt in full (I know they are in the periodic table)?

Thank you for noticing it and bringing it up! We changed it throughout the whole manuscript to not be capitalized.

Line 103: what is a “packed” soil column – not everyone will know it? Incubation would suggest a highly controlled situation, like in an incubation chamber. I would use a different term. “Exposed”, perhaps? What was the agricultural field: arable with cereal crops (wheat and barley) over the two years?

We revised this section (l. 118-121) to make it clearer for a wider audience. Additionally, we altered the wording of “incubated” in this paragraph, though we still believe “incubation” is the appropriate term in other parts of the manuscript where it is used.

Line 106: periodically = after one and two years?

We rephrased the sentence for clarity regarding the time points as follows (l. 123-125):

We utilized X-ray CT to monitor the evolution of soil macropore structure from year 0 to year 1 and year 2, and employed ICP-MS to follow the vertical transport of MPs, both at the one- and two-year time points.

Results

Thank you for presenting summary results as boxplots; they are perhaps more informative than means with errors, BUT to enhance and work towards complete transparency, the raw data are easily transposed on the same graphs. It would be good to start seeing a movement in the scientific community to present the spread (distribution) of the data so that there is no inadvertent hiding of data behind the central tendencies and summarising spread.

From the methods, technical replicates were sampled (n = 3 from layers) but it is not clear what was done with those to calculate the biological replicates.

The following sentence was added at the end of the method section 5.6 to clarify this point (l. 495-497):

After obtaining the concentrations of MPs for each of the three technical replicates per layer, their values were averaged to yield a single mean concentration of MPs per layer for subsequent analysis.

I checked Table S9 and see comments below for the statistical analysis section in the methods about reporting statistical results more transparently. It seems you compared the years, which in an interesting question. However the cores went back into the field each time and are not independent (paired), I would assume. For example, any existing pores in year 0 or 1 would have an effect on the subsequent year(s) whether they remain or disappear – also the df would have changed over time because pots were removed, but only for the treatments. P values of 0 logically not meaningful (absolutely no chance at all), therefore, especially with the sample sizes used, please report as $p < 0.001$ where relevant, but not as 0.000.

Thank you for pointing this out. We reanalyzed the data using linear mixed-effects models (LMMs) to account for the repeated measurements and nested field design. Blocks were included as random effects

(1|Block) to address non-independence across time points (year 0, year 1, and year 2). The updated statistical analysis is reflected in the methods section, and detailed outputs are provided in the supplementary information, with p-values now reported as $p < 0.001$ where appropriate.

I also checked Table S10. It appears that the significance (based on the reported p-values) are mostly at $T = 0$, so straight after the cores were prepared as far as we can tell from the methods. How come that there are significant differences before deployment for a variable which is expected to be the same at $t = 0$ (in macropores in ROI2). If there are inherent differences of response variables between designated experimental units at day dot, then how will that influence subsequent observations?

Thank you for your observation. You are correct that at time 0, the local mixing of MPs affected soil morphology, which likely influenced the initial pore composition. However, the soil structure at time 0 is not our primary focus, as it represents the immediate impact of MP incorporation, in addition to this situation being least realistic since it was a packed soil column at $t=0$. Any initial changes in pore composition at this stage could potentially influence macropore evolution, but our results show that it did not have a significant effect on subsequent observations.

Why is figure 2 highlighted to show only the results from the controls and not reflecting the research aims involving the comparative nature of the experimental treatments?

Thank you for your comment. Since one of the aims of our study was to analyze the overall change(s) in soil morphology over two years, we believe that focusing on the control samples in Figure 2 would make the trend clearer and easier for the reader to interpret. Also, the sample size stays unchanged over the 2-year investigation. While this is a corollary aim of our research, we felt that presenting the control data alone was sufficient to illustrate the temporal changes in soil morphology, without the added complexity of the experimental treatments at this stage of the analysis. Additionally, we chose ROI 3 as it represents the most natural soil, helping to exclude the influence of the artificial sieved layer on the top, which might otherwise confound the interpretation of the soils' natural morphological changes.

I wonder if you could see the pore distribution as multivariate data e.g. the results in line 133 - 137, and thus approach this with a multivariate mind (the pore size profiles can be compared for example, rather than a summarising univariate measure or single size classes). Could be done with an ordination, or permutational techniques. Just a suggestion as it may test hypotheses relevant to changes in pore sizes, but then as a whole.

Thank you for the suggestion. We agree that a multivariate approach, such as ordination or permutational techniques, could provide valuable insights into changes in pore size distributions. However, our current study focuses on summarizing key univariate measures and specific size classes to address the primary research questions. Incorporating a multivariate analysis would go beyond the scope of this work, but we appreciate the idea and may explore it in future studies.

Line 129: I was not aware that the aim was to simulate a freshly prepared seedbed, but also the methods do not really simulate a freshly prepared seedbed (I think that is done differently usually, especially in the field).

It was not the aim to simulate a freshly prepared seedbed but to provide an approximately identical initial structure for all soil samples that resembled a freshly prepared seedbed. The site was prepared by one pass with a rotary tiller followed by one pass with a rotary harrow (mentioned in the method section 5.3). We would argue that preparing a field with a rotary tiller followed by a rotary harrow is a typical method of seedbed preparation, especially in field settings. This approach is commonly used to create a fine, workable soil structure that promotes good seed-to-soil contact and supports successful germination. Often, the soil surface is slightly compacted after sowing using a roller, resembling the way we packed the topmost 2 cm of the soil.

We changed the according text (l. 157-159) to emphasize the similarity without overstating it to a seedbed and to give a better rational why we used ROI 3 only for the analysis:

Using our dataset from the control columns (no MPs), we investigated the temporal evolution of soil morphologies in ROI 3, where we had an initial soil structure resembling a freshly prepared seedbed by avoiding the less natural sieved soil layer on top (ROI 2).

Line 128 - 143: Is this section describing results for ROI3 in the controls only, given that this refers mostly to figure 2? It is not clear why there is such an emphasis on this bit of the controls only which is not really reflecting the aim of unravelling “complex interactions” of microplastics that well.

Indeed, those lines described the results for ROI 3 only in the control columns. As previously mentioned in the discussion of Figure 2, one of the key aims of our study was to analyze the overall temporal changes in soil morphology over a two-year period. We focused on the control columns, specifically ROI 3, because it provided a clear and easily interpretable representation of these changes, without the added complexity of MPs treatments. By highlighting the controls, we were able to demonstrate the natural progression of soil morphology in a more straightforward manner. Additionally, ROI 3 was selected because it represented the most natural soil structure, avoiding the confounding effects of the artificial sieved layer. While the comparative nature of the experimental treatments is a central aspect of our study, we felt that for this analysis, focusing on the controls alone provided the clearest insight into the temporal changes in soil morphology. Moreover, since there were little to no differences between the control and MPs treatments in this context, the choice of which ones to display does not affect the results.

Is Figure 3 also the controls only? I assume these are some key examples of the images and do not represent the full set of results?

Figure 3 displays results from the control columns, but it represents a set which is representative of the $n = 15$ replicates. Of course, there is variability in the data which is also represented in Figure 2e and 2g in the manuscript. By looking at Figure 2e and 2g it also supports the fact that biopore volume increased over time while POM volume fraction was overall more similar with a small decrease over time.

Below, we provide an additional visual overview of another control sample (column 34) over the two-year period, showing biopores and POM representation. We created this 3D representation specifically for the reviewer to offer further insight into the control data. While we believe this visually aids the readers' understanding, we do not feel it is necessary to include more 3D representations in the manuscript or supplementary information, as the boxplots more effectively capture and convey the overall trends.

Figure 1 for Review: Image resolvable biopores (top row, white) and POM (bottom row, green) of ROI 1 for the three yearly sampling timepoints. Panels (a), (b), and (c) correspond to biopores at year 0, year 1, and year 2, respectively. Panels (d), (e), and (f) correspond to POM at year 0, year 1, and year 2, respectively

Line 202 and 205: is it also notable when the top layer contained all the MPs to begin with? Also, obviously it was the “highest” because it was the *top* layer; “greatest” or “most” maybe?

We revised the wording to use 'greatest,' as the reviewer suggested, and clarified that the highest concentration in the uppermost layer was expected, given that MPs were initially spiked there, so as not to imply any surprise at this finding.

Line 206 - 208: this is discussing the results, and it probably should go to the discussion section where you can focus on explaining these findings more, supported by scientific literature.

You are absolutely right. In response to this suggestion by the review, we have added the specified lines along with an additional sentence in the discussion section as shown below. However, we don't believe there is relevant literature that directly supports this statement, so we have focused on interpreting our data.

See new lines 319-322:

Still a considerably high concentration of MPs was observed in the second layer. This may be attributed to short-distance transport of MPs, but also to spatially heterogeneous soil settling resulting in an uneven interface between the MPs-spiked soil layer and the layer below.

Line 210: are you referring to all layers you sampled for the observation that the LOD was exceeded?

Yes, we refer to all layers that had higher values than the MDL. We adapted the line (l. 249-251) as follows:

Notably, measured MPs concentrations in all sampled soil layers exceeded the method detection limit (MDL) of 1.52 mg/100 g soil.

Line 212 Would you agree that losing from the system about 16% of the fragments you added to the system is maybe something to worry about? Quick maths (please double check): you added 1.3 gram of the plastics to a core (see line 362), so at a recovery ratio of 83.4% (the lowest recovery, after 2 years) there was a loss of $0.16 * 1.3 = 0.208$ gram. Scaling that up it would be 26 gram m^{-2} of fibres and / or plastics, which I would argue is quite substantial. Is this a technical loss due to measurements and rounding (but within the accepted error margins), or can you find out that these particles did not end up somewhere else in the field?

We agree that a certain fraction of MPs might be lost from the columns and therefore be in the surrounding environment in the field. According to our calculations, the average loss for the fragments and fibers would be 12.4 % and 9.8% after two years, respectively, by also including the uncertainty from the recovery method itself. We have added a brief note addressing this loss in the results section 3.3 (l. 335). Consequently, according to these calculations it could be an average loss of fibers of 160 mg for fragments or 127mg of fibers per column. This is a slightly smaller loss as calculated by the reviewer. Still this amount should be accounted for. We see several factors that might have led to this loss from the recovered MPs, including:

- i. For cutting the frozen soil column cores, we used a stone saw blade that has a thickness of 2 mm. This contributed to a loss of 8.8 % from the overall soil material ($100 \% * 4 \text{ cuts} * 2 \text{ mm} / 90 \text{ mm} = 8.8\%$). Of course, this may not sum up to 8.8% of the plastic, since it is not evenly distributed in the column. An estimate of the effective loss of MPs from this source might be $\sim 3 \%$ by accounting for the gradual MPs distribution in the column.
- ii. Another source of loss could arise from the transport by earthworms (or other soil biota) outside of the column both horizontally and vertically. However, this portion of the loss cannot be accurately estimated. One approach to investigating this further would be to identify earthworm burrows and examine those that pass through the column walls or extend out from the bottom. However, this would warrant a separate study in itself.
- iii. Further, the soil stuck in the holes of the column were not used for the analysis and therefore could also have led to a loss of MPs that were localized in this part. Per column roughly 35g dry soil was in those holes.
- iv. Erosion from the soil surface: MPs could also be lost through wind and splash erosion from the soil surface. This mechanism, while plausible, could not be quantified in this study but has been noted as a potential factor contributing to the loss.

We added few lines that mention the factors leading to the potential sources of this MPs loss either during analysis or hypothetical losses due to soil fauna and other natural conditions in the field experiment. See lines 335-340:

*While a portion of the MP loss can be attributed to the analytical process, such as soil material removed by the saw blade during soil core cutting and soil retained in column holes that was excluded from analysis, this alone does not fully explain the reduction in MPs recovery rates observed. It is likely that additional losses were due to MPs migrating vertically within the soil, being redistributed or laterally transported by soil fauna and roots, or being lost from the soil surface through wind and splash erosion.*¹⁷⁻²⁰

Finally, we would like to emphasize that a recovery rate of approximately 85% in a field setup with crops, exposed to weather over two years, is considered a strong result. While we refrain from self-congratulation, we believe this outcome reflects the robustness of our method in realistic field conditions.

Figure 5: I wonder if you can place these depth profiles side by side based on year? It looks like the x-axis is log scale, so that distorts the boxes (IQR) and the whiskers. Can you also state this in the caption?

We added the information about using log-scale as x-axis in the figure caption and it reads as follows:

*Figure 5: Vertical transport of PET MPs fragments (cyan) and fibers (violet) after one- (solid bars) and two- (hashed bars) year incubation in the field. The x-axis is presented on a log scale, which compresses higher values and can affect the proportional appearance of the boxplots. Method detection limit (MDL) is shown as black dashed vertical line. In the box plots, line: median, box: lower and upper quartile, whisker: highest and lowest value, empty dots: outlier. Statistical significance for effect of replicate is indicated as follows: p-value <0.001 ***, <0.01 **, and <0.05 *.*

Discussion

The first part of the discussion (221-242) does not seem to reflect much on the overall aims of the study regarding interactions with microplastics. Instead, this section covers mostly general soil physics, which of course is important as a foundation, but could be brought in light of the experiment and its setting more.

The discussion of the pore-network evolution in the control samples sets the baseline of how the soil properties associated with the soil structure develop without the presence of the microplastic. It appears logical to us that such a section needs to be included in the manuscript. We therefore are not convinced that this section requires more explicit linkage to the MPs aims of the study.

Line 221: Be careful not to cherry-pick the significant findings only. However, why can't we explore this finding more that the MPs are not causing any pore changes (which is for better or worse) I would argue any deviation from the control is something to explore further, even if it is not significant internally for a treatment.

Thank you for your thoughtful comment. We agree that any deviations from the control, even if not statistically significant, are worth further exploration. To address this, we not only examined statistically significant findings but also considered point estimates and included results with large p-values, acknowledging the uncertainty surrounding these outcomes. While the evidence for some effects is uncertain, we believe it is important to mention these observations as they could still provide valuable insights. Additionally, we have adopted the phrasing in the results section (as suggested by Reviewer 2 from this source: "<https://reproductive-health-journal.biomedcentral.com/articles/10.1186/s12978-021-01131-w>") to better present the findings, including these uncertain results, so that readers can fully appreciate the context and variability in the data.

Line 231: was there any evidence that there were earthworms in your soils (and more importantly the cores)? Otherwise this explanation, why valid in general, does not hold well for the observed results. Furthermore, the cores did contain three plants with their roots, including after being harvested and (I assume) left to decompose.

Please see our previous reply to a similar question. Regarding the plant roots, indeed, they were left to decompose after harvest.

Line 244 – 262: it is quite curious to see a large section in the discussion such as this with no support from the scientific literature – unless I missed something, I see no citations at all here. Please add relevant citations to this section, particularly to key statements which certainly need it (e.g. line 253, but there are many more, particularly line 256 where you curiously mention leached metals, which you obviously added to the plastics as well...). Without proper support from relevant scientific evidence, this will be considered poor form I am afraid.

We agree that more literature should be used to support the discussion. Therefore, we added relevant literature as reference directly in the manuscript:

- Cao, X.; Liang, Y.; Jiang, J.; Mo, A.; He, D. Organic Additives in Agricultural Plastics and Their Impacts on Soil Ecosystems: Compared with Conventional and Biodegradable Plastics. *TrAC Trends in Analytical Chemistry* **2023**, *166*, 117212. <https://doi.org/10.1016/j.trac.2023.117212>.
- Guo, J.-J.; Huang, X.-P.; Xiang, L.; Wang, Y.-Z.; Li, Y.-W.; Li, H.; Cai, Q.-Y.; Mo, C.-H.; Wong, M.-H. Source, Migration and Toxicology of Microplastics in Soil. *Environment International* **2020**, *137*, 105263. <https://doi.org/10.1016/j.envint.2019.105263>.
- Chae, Y.; An, Y.-J. Current Research Trends on Plastic Pollution and Ecological Impacts on the Soil Ecosystem: A Review. *Environmental Pollution* **2018**, *240*, 387–395. <https://doi.org/10.1016/j.envpol.2018.05.008>.
- Zhu, D.; Chen, Q.-L.; An, X.-L.; Yang, X.-R.; Christie, P.; Ke, X.; Wu, L.-H.; Zhu, Y.-G. Exposure of Soil Collembolans to Microplastics Perturbs Their Gut Microbiota and Alters Their Isotopic Composition. *Soil Biology and Biochemistry* **2018**, *116*, 302–310. <https://doi.org/10.1016/j.soilbio.2017.10.027>.

Line 245: I don't think you explicitly measured burrowing nor its behaviour, at least it was not clear from the methods. The results only show pores at different time points, but it is an assumption that these are caused by burrowing. Please be careful with interpretations like this.

You are correct; we did not explicitly address worm burrows, or the behavior associated with earthworm activity. To avoid confusion, we have revised the sentence by removing the phrase “*and its potential impact on burrowing behavior and root growth*”.

Line 246: ah now I can see that you consider the spiking is considered high and there is something about highly contaminated sites. There is still no citation for this, nor much of a justification to add that level of MPs to an arable soil.

adapted the method section (lines 428-433):

The 1 wt.% MPs concentration was chosen to represent a higher contamination scenario, reflecting potential hotspots in areas with significant plastic pollution, while still being within the range observed in some industrially impacted regions²¹⁻²³ This level helps assess the impacts of elevated plastic concentrations in environments with intense agricultural or urban plastic use and was also selected to ensure concentrations were high enough to be reliably measured in the soil matrix by our method (ICP-MS).

And adapted the following lines (l. 289-292) in the discussion section:

Contrary to the initial concerns regarding significant alterations in soil structure due to the presence of MPs, our study reveals that changes are minimal and transient even at high MPs concentration of 1 wt.%, which may be relevant for hotspots of contamination or regions with cumulative plastic application over time.

Line 264: the presence of indium in the soil was shown, which you can assume was representative of the microplastics (slight difference!).

We adapted the sentence so it is clearer that the MPs were measured via the indium as shown below (l. 309-310):

The vertical transport of MPs was conclusively shown by measuring indium, which correlated with the MPs. Both MPs fragments and fibers exhibited a consistent trend across the two-year monitoring period.

Line 270: do you think that another similar experiment but without plants would help test part of that hypothesis? Highest = greatest or most?

We assume reviewer refers in the first question to the effect on biopores and in the second one to the “highest concentration of MPs”. Indeed, an experiment only looking at the effect of bioturbation without the impact of root growth (and/or soil macrofauna) would be a good addition, but this was out of scope of our current possibilities because of the time effort involved in expanding the experimental matrix to such a large extent. Nevertheless, with this approach we would be able to better disentangle bioturbation from plant growth/root growth. Although, having a field experiment where there is no root growth happening is very difficult to achieve and would call for application of herbicides, which we didn’t want to include in our study. Keeping macrofauna out would have been even more challenging, requiring packing the samples into sufficiently fine-meshed meshed material. To improve clarity, we have revised the wording to 'greatest,' as it more accurately reflects the intensity of the concentrations.

Line 273: not sure where the nanoplastics come into play here, it seems to be a distraction.

Thank you for the feedback. We have removed the mention of nanoplastics from the sentence to maintain focus and avoid potential distraction.

Line 276 - 278: it is also worth considering that [39] employed *Lumbricus terrestris* a large worm, which has quite a distinct ecological niche in that it burrows deeply (up to metres deep), lives in permanent burrows and pulls in decaying material to ferment in the burrow. I always wonder how representative this species is for pot studies, even done in a field (but that is irrelevant now) – however, there are probably more processes in the soil than just a big burrower, and you would have seen evidence of such activity on the surface (I think it is likely that a single *L. terrestris* would have a complete 706 mL soil core for breakfast, if they were there). I would recommend using different evidence to support your hypothesis for this phenomenon.

We can appreciate the reviewers’ concern that different species of earthworms will have different behaviors and ecological functions. There is a very nice recent publication by Capowiez et al. demonstrating this.²⁴ We believe this reference is still valid to include in this instance as *L. Terrestris* is an important species in Swiss soils and has been found in surveys at neighboring fields. It is therefore highly likely that *L. Terrestris* has shaped a considerable part of the earthworm burrows imaged in our soil samples. This is also relevant, because *L. terrestris* is a prime candidate for an earthworm that transports MPs to deeper soil depths, due to the burrowing behavior described by the reviewer. *L. terrestris* is also a well-studied organism used across many experiments. Since there is a limited amount of information on how other worm species interact with and/or redistribute plastics specifically, we believe this reference can still give readers and impression of the processes which may be happening in the environmental system. However, we have added the following line to the manuscript in order to make it clear that *Lumbricus* are not the only organisms which can have an impact on MPs bioturbation and that there are other earthworms with different burrowing behaviour:

Added lines 328-330:

*It should be noted that the burrowing behavior of different earthworm species and other soil-dwelling organisms can deviate from the one of *L. Terrestris*.²⁴ It follows that also rates and directions of material redistribution could also differ.*

Line 280 - 286: this explanation for the loss of MPs will also need support with relevant science, please.

We revised the text to more clearly address losses due to analytical and technical factors and added relevant references to support the potential losses caused by soil fauna, roots, and surface losses from wind and splash erosion. See new lines (335-340):

While a portion of the MP loss can be attributed to the analytical process, such as soil material removed by the saw blade during soil core cutting and soil retained in column holes that was excluded from analysis, this alone does not fully explain the reduction in MP recovery rates observed. It is likely that additional losses are due to MPs migrating vertically within the soil, being redistributed or laterally

transported by soil fauna and roots, or being lost from the soil surface through wind and splash erosion.^{17-20,25}

Conclusion and implications:

Line 288 – 293: this part is introductory and not based on the results, the conclusions and implications should be based on the findings presented in this study.

Thank you for pointing this out. We removed the introductory lines to directly focus on the findings and results of our study.

The rest of the conclusion section is largely a summary, instead it would be better to highlight the importance of the findings. The last part looks good about wider implications, but the use of alternative plastics as a recommendation needs much more research before they are to be deployed without testing and regulation.

We have revised the conclusion and implications section to better highlight the significance of our findings and their potential implications. The following lines have been added:

See lines (348-350):

This slow migration suggests that MPs could accumulate in the upper layers of soil, potentially creating localized pollution hotspots, while also offering an opportunity for targeted remediation efforts.

See lines (354-360):

This suggests that soil properties of seedbeds evolve towards a more settled and naturally structured soil with minor influence of the presence of MPs. Our study provides reassurance that MPs, at least at concentrations of 1 % wt., may not disrupt soil structure formation and development as much as initially feared. However, higher MP concentrations in hotspots could still affect soil macropore networks and other critical soil properties, emphasizing the need for continued monitoring and research on the long-term effects of MPs on various soil properties.

See lines (369-373):

In light of these findings, reducing plastic use in agriculture and exploring sustainable alternatives are important goals. However, alternatives to plastics should undergo thorough testing and regulation before widespread adoption to ensure they do not introduce similar or new risks to soil ecosystems. By prioritizing sustainable solutions, we can mitigate the potential for irreversible soil pollution, reduce plastic-related ecological footprints and better safeguard agricultural ecosystems.

Materials and Methods:

Line 315: I checked (on the 26th of September 2024) the online catalogue of the company for this type of PET and I can't seem to find the particular product you refer to. I did find PET "yarns", which I suspect were used, but what else did these plastics contain other than PET – did you check? Can this be clarified, perhaps you already did so in your work presented in [51] and [52], or elsewhere?

Thank you for your comment. Unfortunately, we do not have further detailed information about the specific composition of the PET material at hand. However, PET is typically a relatively pure polymer with only small amounts of additives used in its manufacturing process.

Line 323-234: this is a bit confusing, are these sentences double per chance? Also I can see from source [51] that fibres were produced, and cut at around 500 um. What are the difference from the current methods compared to that described in your work presented in [51] and [52]? Careful with self-plagiarism, I know that is difficult for repeated methods though.

We added the references after the method description for fragment and fiber production, to avoid self-plagiarism. Further, we changed the lines so that it is clear that the metal doped pellets were produced

as described in Frehland et al., see line (384-387):

The production of the indium doped PET fibers was performed according to the methods described in Frehland et al.²⁶, and the indium doped PET fragments were produced following the procedures detailed by Tophinke et al.²⁷

Line 351: how many 2 cm holes were there and were they covered somehow to keep soil inside (if needed)? Consider referring to FigS10?

Each column had 24 perforations with a diameter of 2 cm. Although we initially considered covering the holes, it turned out to be not needed, as the soil remained stable inside the columns and did not fall out of the holes, even during transport for further analysis in the laboratory. We clarified the number of holes per column directly in the manuscript (l. 416) and refer the reviewer to Figure S10 in the SI (l. 418).

The cores were about 706 mL, with a depth of 9 cm, which could be considered rather confined for relevant mesofauna such as earthworms (typical pore formers) and even plant roots. I understand that larger cores would significantly increase the cost, but the relatively small size of the physical experimental units should also be considered when extrapolating the results.

Larger columns would not only increase the costs but also worsen the image resolution. The chosen column size represents a compromise. Please see the explanation given to a previous question. It is a central challenge of soil science to bridge the scale between pores and landscape, as many important processes are governed at the pore scale but knowledge on their effect is desired at the field or landscape scale.

Line 362: you may have expected this question: what is the rationale to spike the top soil layer at 1% w/w it is quite... a lot? I understand that for the cores as a whole the density is less, but that was not done.

There are two main reasons for this decision:

- i. The reported microplastic concentration in soils spans from 0.005 wt % up to 6.7 wt % and is largely dependent on the location and industrialization of the site.^{21,22} Kalberer et al. performed a material flow analysis in the context of Swiss agriculture and estimated the annual amount of plastics applied to agricultural land in Switzerland. Through cumulative application over the years, the authors found that this can lead to a maximum concentration of up to $0.02 \pm 0.01\%$ ($200 \pm 100 \text{ mg kg}^{-1}$) of plastics in agricultural soil.²³ In our experiments, we added 1 wt % microplastic in the top 2 cm of the soil column, which corresponds to a concentration of microplastic that lies within the above mentioned environmentally occurring range. While this is higher than what was estimated specifically in the Swiss context, our objectives aim to also understand impacts of plastics on a broader scale outside Switzerland.
- ii. To reliably detect microplastics in the soil, we aimed to achieve a final concentration that would allow us to detect at least 1% of the initially added MPs in a given soil layer, which is equivalent to 13 mg of MPs per layer. To meet this requirement, we decided to spike the top soil layer with a total of 1% by weight of MPs, which corresponds to 0.22% MPs by weight in each column. This approach ensured that we could confidently detect at least 1% MPs in each soil layer. While it would have been possible to use a lower concentration, such as 0.22% in the top layer or 0.048% overall, this would bring us closer to the method's detection limit, potentially compromising the accuracy and reliability of our results.

Regarding the reviewer's comment, "I understand that for the cores as a whole the density is less, but that was not done," we were unable to fully understand the intended meaning or suggestion for improving. We apologize for any confusion.

I know that mixing plastics homogeneously into soil, particularly fibres, can be very challenging, and fibres at 1% can be very 'fluffy'. However, the mechanical mixing of the plastics in the 2 cm layer would very likely have changed the pore composition and soil structure (and bulk density) compared to the rest which you let sit for 5 days to settle, even after "delicately" compression. Can you comment/clarify how the vertical continuation of the physical composition was ensured?

Indeed, mixing fibers in soil requires an initial separation of the fibers so that they are not clumping together. For the experiment, all layers on top of the columns were sieved soil, to which we added either 1.3 g of MPs or 1.3 g of additional sieved soil. While adding MPs instead of soil may slightly affect the density of the top layer, the change is marginal.

Before adding the sieved soil layers, we filled all columns with field soil to the same volume (fully packed, without compression) and allowed it to consolidate for 5 days. The top 2 cm of this soil were then removed and replaced with the sieved soil (either with or without MPs). To prepare the MPs mixture, we added 130 g of sieved dry soil with 1% MPs (1.3 g) into Schott bottles and mixed it for 5 minutes in a Turbula® mixer. Afterward, we poured the mixture into a metal ring and added 35 mL of water to help bind the soil and MPs together. This ensures homogeneity in the mixture. Next, the wet soil mixture was compressed in a pipe (matching the diameter of the columns) using a plunger and gravity. Finally, the compressed mixture was carefully added to the top of the columns. All top layers should have the same bulk density (0.82 g/cm³). The estimated overall bulk density of the whole soil cores after year 1 was 0.86 ± 0.05 g/cm³ and after year 2 was 0.98 ± 0.07 g/cm³.

Soil Bulk Densities Estimations:

$$\text{Bulk density} = \frac{\text{Summed up weight of layers}}{\pi \cdot \left(\frac{\text{Diameter}}{2}\right)^2 \cdot (\text{Height} - \text{Height of cuts})}$$

$$\text{Bulk density} = \frac{\text{Summed up weight of layers}}{\pi \cdot \left(\frac{10 \text{ cm}}{2}\right)^2 \cdot (9 \text{ cm} - 4 \cdot 0.2 \text{ cm})}$$

→ Year 1: 0.86 ± 0.05 g/cm³ (n=10)

→ Year 2: 0.98 ± 0.07 g/cm³ (n=20)

We added a line in the results section that shows the evolution of the soil bulk density over time. See new lines (270-272):

Consistent with this trend, the estimated bulk density of the whole soil column increased over time, from 0.86 ± 0.05 g/cm³ after year 1 to 0.98 ± 0.07 g/cm³ after year 2.

Line 371: "reinserted" - I understood that the samples were kept in the lab until this point?

We updated several lines in the manuscript during revision, so it should be clearer now that the samples were first put in the field to consolidate, then brought to the lab to add the MPs-spiked soil layer on top, and then stored in the cold room until the first X-ray scans were performed. After that, columns were reinserted into the field.

See lines 438-439:

All columns were stored in a dark cold room at 4° C prior to transportation and X-ray scanning.

And line 441:

Samples were kept in the dark cold room at 4° C until X-ray scans were conducted (timepoint: year 0).

Line 372: With such small surface area (~79 cm²) even three seedlings are quite high (382 seeds m⁻²). Can you comment on how this represents a typical sowing density? There are three big, maturing wheat plants growing in a 706 mL container (ok, it is in the soil) so how is that representative of 'the rest of the field', and then there will be barley the next year.

Please see our reply to a previous similar question.

Did the wheat seeds germinate in 7 days and all of them were in the pots were viable? Were there any roots in the soil which you had to remove or left (these would be pores, albeit probably small)?

The wheat seeds had a longer time to germinate, since they were added on the 15. October together with the 2 cm thick sieved soil layers. Although columns were stored in a dark cold room, the seeds could still germinate. We reduced to 3 viable plants in the initial phase over few months to ensure only 3 plants could grow. In case there were roots formed, we left them to decompose in-situ to not interfere with the soil or disrupt the MPs distribution within the soil layers.

Line 372: Figure S9 is helpful, but it only shows 4 x MP fragments pots being sampled, and 5 x MP fibres, can you please check?

Very attentive! Thank you for pointing it out. It is of course 5 of both treatments that were sacrificed. We adapted figure S9 accordingly.

Line 374: how did you harvest the wheat? By hand with a tractor? Did you leave the stubble?

We adapted the text the following to explain how we harvested the wheat (l. 444-448):

We harvested the winter wheat on July 11th, 2022. The wheat grown on the soil columns was harvested by hand, leaving a stubble of roughly 1 cm, while the wheat from the rest of the field was harvested using a bar mower, after the columns had already been removed from the field. All plants that had matured on a soil column were dried for further analysis to assess the impacts of MPs on the crop yield.

We also clarified it for barley in lines 456-457:

Barley was harvested on June 22nd, 2023, using the same method as for the wheat, and all plant material collected from the soil columns was dried.

Line 376: ah I see that initial measurements were at n = 15 (wheat), and some of those were done at n = 5 (due to destructive sampling) leaving the subsequent experimental work at n = 10 (barley), but not the controls which were left at n = 15. Is that correct and if so, can you clarify that?

Exactly like that. We updated the text so it will be clearer (l. 450-453):

Subsequently, five columns of each MPs treatment (fibers and fragments) were randomly selected and sacrificed for further analysis to assess the vertical transport of MPs. Due to the destructive nature of the vertical transport analysis, the sample matrix was reduced to 10 columns containing MP fibers, 10 columns containing MP fragments, and 15 control columns.

What happened with the remaining cores while waiting to be inserted back into the field after the wheat harvest? Can you explain storage conditions?

All cores were wrapped in aluminum foil and brought to a dark cold room at 4° C where they were

stored until analysis was done and brought back to the field.

Following lines (l. 461-462) were inserted to explain storage conditions:

Whenever the columns were removed from the field for analysis, they were wrapped in aluminum foil and stored in a dark cold room at 4°C.

There was no addition of any fertiliser in between crop rotations - can you specify?

We added line (463-464) if necessary, we can specify the amount and type of fertilizer. However, we are convinced that this information is not relevant for our study:

The experimental field was fertilized according to agronomical best practice throughout the duration of the experiment.

Line 372: Figure S10 is really helpful! It would also have been good to include some universal item to show scale. It does not show crop “progression” though, there is only an overview of the field showing that the wheat was there.

Thank you! We included some scale by adding the most important lengths of the field, distance between columns and diameter of the columns directly in the images. We decided to not show crop progression, since it seemed not very informative to us.

Line 388: can you specify how many soil columns? N = 5 for wheat and n = 10 for barley and n = 15 for control?

We specified the number of replicates for wheat and barley in corresponding lines 466-470:

All soil columns were stored and maintained at -20°C. The soil columns containing MPs (n = 5 for both fibers and fragments with wheat in year 1, and n = 10 for both fibers and fragments with barley in year 2) were carefully removed from the aluminum columns and sectioned into five layers using a stone saw, each approximately 2 cm thick, except for the bottom layer, which was variable between 0 - 1 cm due to uneven compaction over the experimental time.

Line 391: with a high risk of cross contamination of indium, can you clarify what you did to prevent this (I would imagine dust and adsorption to equipment)?

The lab equipment such as pestle and mortar, and sieve were rinsed with water and ethanol and dried after processing each layer. Additionally, we began processing with the lowest layer (layer 5) and proceeded sequentially to the top layer, where we expected the highest concentrations of MPs and indium.

see new line (472-475):

To prevent cross-contamination of MPs between soil layers during analysis, the lab equipment was rinsed with water and ethanol and dried after processing each layer. We began processing with the lowest layer (layer 5) and worked sequentially to the top layer, where the highest concentrations of MPs and indium were expected.

Line 394: Quickly check if I still get it: the cores were 9 cm tall, so with the error margin of 0-1 cm and layers of 2 cm, there were $8/2=4$ layers to consider of 2 cm and the deepest layer was there at 1 cm or not at all? Wheat had n = 5 for the treatments, so there were 10 pots in total, which were sliced into 4 or 5 layers, so 40 or 50 layers in total. Each layer had a technical replicate of n = 3, so there were 120 or 150 samples for ICP-MS. For barley this was double, because there were 10 pots left for the treatments, so 260 – 300 samples digested for ICP-MS. Is this correct? If not, then can you please make sure this is crystal clear?

Yes, that is correct. For all analyzed columns, there was only one column containing fibers which was analyzed after 1 year when wheat was crop that had only 4 layers. All the others had 5 layers as specified. And indeed, we had 147 ICP-MS measurements after year 1 and 300 ICP-MS measurements after year 2.

What was done with the technical replicates – were they averaged?

We added new lines to clarify that point, see lines 495-497:

After obtaining the concentrations of MPs for each of the three technical replicates per layer, their values were averaged to yield a single mean concentration of MPs per layer for subsequent analysis.

Line 410: we would really like to know the “known” In-to-plastic ratio, because we don’t know it. Particularly relevant would be how you made that standard curve, ensured accuracy and groundtruthed it. I know that relative errors are acceptable comparing *across* treatments (effect sizes) if these errors are kept consistent and everything else is the same, but the aim here is to reflect back on actual MPs in the soil layers. That would mean that we really would like to know the In-to-plastic ratio of the spiked soil too. Can you address this please?

We made a calibration where we added either 10 mg, 20 mg or 30 mg of MPs (in triplicates) to 500 mg soil and digested it via microwave acid digestion. Then we measured the indium concentration via ICP-MS. The PET fragments showed an In concentration of 0.198 ± 0.006 wt.% and PET fibers an In concentration of 0.186 ± 0.008 wt.%. The indium background concentration of 1 g dry digested soil was $0.058 \mu\text{g/L} \pm 0.004 \mu\text{g/L}$.

We added the following lines to clarify the In-to-plastic ratio (lines 491-493):

MPs concentrations were then derived from the measured In concentrations using the experimentally defined In-to-plastic ratio (0.191 ± 0.004 wt.% for PET fragments and 0.185 ± 0.003 wt.% for PET fibers).

Line 431: Great to see the check for leaching. Assuming that all of the added plastics remained in the mesh bags during the two years, would it have been useful to see if In was detectable in the soil directly under that bag?

We are confident that not much indium leached out from the MPs as earlier tests of the MPs in harsh environments such as gastrointestinal solution led to very minor indium leaching. While measuring the soil surrounding the mesh bags would have been beneficial, we doubt it would have been feasible to detect the extremely low indium concentrations in the surrounding soil. Moreover, the mesh bags had a mesh size of 30 μm .

Line 432 - 434: Are those values (in wt%) within the credible detection limit and error range of the equipment? Could do a quick and easy paired t-test, or similar test if the data are not parametric, to show that there was or was no significant decrease.

Based on sample digestions yielding roughly 10 mg of MPs, the indium concentration was approximately $9.7 \mu\text{g/L}$, substantially above the method detection limit of $0.071 \mu\text{g/L}$. This confirms that the detected values are within the measurable range of the equipment.

Given that the differences in concentration are very small between initial indium concentration and after 2 years incubation in the mesh bags (on the order of just a few percent), a statistical test such as a paired t-test may not be necessary. The values are within the expected measurement precision of the ICP-MS, and the observed decrease falls well within the margin of error. Therefore, a statistical test would likely not provide additional meaningful insight in this case.

Line 439: here and elsewhere, can you specify the level of replication you employed for each measurement?

We assume that the reviewer is referring to the number of columns that were scanned. Therefore, we added the lines below to the manuscript to clarify the replicate numbers for each measurement.

Lines (533-535):

The measurements at year 0 were performed with 15 replicates for fiber columns, 15 for fragment columns, and 15 for control columns.

Lines (538-540):

At year 1, the measurements were performed with the same replication as year 0: fragments ($n = 15$), fibers ($n = 15$), and controls ($n = 15$). At year 2, the measurements included fragments ($n = 10$), fibers ($n = 10$), and controls ($n = 15$).

Line 498: which version of R was loaded in RStudio for the analysis? Could you also include references to provide credit to the developers where it is due?

Added line 599:

Statistical analysis was performed using RStudio 2022.07.1, with R version 4.4.2.

Line 501: for objective 2) analysis over time, how did you deal with the non-independence of the samples and how did you deal with the unequal replication? What do you mean with “grouping the different treatments” please explain.

As similar concerns have been raised previously, we refer the reviewer to the other comments addressing these issues. However, to clarify, the entire statistical analysis section has been revised. In the updated version, we specifically addressed the non-independence of the samples and unequal replication by employing linear mixed models. These models account for the hierarchical structure of the data and allow us to properly handle the dependencies between samples as well as the unequal replication across treatments.

The one-way ANOVA really only works when you have means which are independent. It is not clear what the independent units are given the many steps and subsamples. You also potentially have $n = 5$, $n = 10$ and $n = 15$ so that will affect the degrees of freedom to calculate relevant probabilities from the test statistic. I thought the experimental design was nested (see line 371) at the biological (field) level, so how was the nesting taken into account in the stats?

Please see comment above regarding the new statistical analysis.

Welch’s tests use estimated degrees of freedom instead of the actual ones, can you explain how that works where you already have (I think!) very different degrees of freedom to start with (particularly when comparing across time, but also barley which appears to be $n = 15$ for control and $n = 10$ for treatments)?

Thank you for pointing that out. As the statistical analysis has been revised and updated, there is no need to address that specific comment further. Please refer to the comment above for details on the revised statistical approach.

Line 508: did you use a package for the Dunn test? I am not aware if native R has that test. If you use any packages, please cite the authors to give credit where it is due.

Thank you for pointing that out. As the statistical analysis has been revised and updated, there is no need to address that specific comment further.

Line 509-511: I am afraid that p-values on their own are very uninformative. I also think that we should step away from the arbitrary reporting of p-values without any supporting information of how they are calculated. The minimum information for an ANOVA would be the F statistic, associated within (treatment) and between (residuals) group degrees of freedom. The treatment df for your design should be 2 in a one-way ANOVA, but the residuals are very dependent on how many replicates you have in total - and that is not made clear in the methods. Without these reported clearly, readers cannot double check the significance and the relevance of the tests, and have to take your word for it but it lacks transparency. Given that you also may have to consider the nestedness of your design in the field, you may need to calculate the F statistic based on the MSS and degrees of freedom of the random nesting factor usually a blocking factor in the traditional agronomy literature, but it is not clear that this was considered.

We appreciate your points about the reporting of p-values and the importance of transparency in statistical analyses. We address these concerns and revised the statistical analysis and used linear mixed models (LMM) to account for both fixed and random effects, as outlined in the revised methods section 5.10. We now explicitly report the F-statistics, degrees of freedom for both the treatment and residuals, and associated p-values for each of the main factors, as well as their interactions. The random effects and their variance components (e.g., blocks and residuals) were also considered to account for the nested structure of the data (15 blocks/nests). The results are summarized in both the ANOVA and post-hoc tables in section S11 in supplementary information. We hope this provides clarity on the statistical approach and enhances transparency in the reporting of the results.

References

[21] needs to state author name(s).

We updated the reference accordingly, so it shows now authors name:

Rillig, M. C.; Lehmann, A. Microplastic in Terrestrial Ecosystems. *Science* **2020**, *368* (6498), 1430–1431. <https://doi.org/10.1126/science.abb5979>.

References

- (1) Sa'adu, I.; Farsang, A. Plastic Contamination in Agricultural Soils: A Review. *Environmental Sciences Europe* **2023**, *35* (1), 13. <https://doi.org/10.1186/s12302-023-00720-9>.
- (2) He, P.; Chen, L.; Shao, L.; Zhang, H.; Lü, F. Municipal Solid Waste (MSW) Landfill: A Source of Microplastics? -Evidence of Microplastics in Landfill Leachate. *Water Research* **2019**, *159*, 38–45. <https://doi.org/10.1016/j.watres.2019.04.060>.
- (3) Elimelech, M.; O'Melia, C. R. Kinetics of Deposition of Colloidal Particles in Porous Media. *Environ. Sci. Technol.* **1990**, *24* (10), 1528–1536. <https://doi.org/10.1021/es00080a012>.
- (4) Kretzschmar, R.; Barmettler, K.; Grolimund, D.; Yan, Y.; Borkovec, M.; Sticher, H. Experimental Determination of Colloid Deposition Rates and Collision Efficiencies in Natural Porous Media. *Water Resources Research* **1997**, *33* (5), 1129–1137. <https://doi.org/10.1029/97WR00298>.
- (5) Elimelech, M. Effect of Particle Size on the Kinetics of Particle Deposition under Attractive Double Layer Interactions. *Journal of Colloid and Interface Science* **1994**, *164* (1), 190–199. <https://doi.org/10.1006/jcis.1994.1157>.
- (6) Keller, A. S.; Jimenez-Martinez, J.; Mitrano, D. M. Transport of Nano- and Microplastic through Unsaturated Porous Media from Sewage Sludge Application. *Environ. Sci. Technol.* **2020**, *54* (2), 911–920. <https://doi.org/10.1021/acs.est.9b06483>.
- (7) Pradel, A.; Delouche, N.; Gigault, J.; Tabuteau, H. Role of Ripening in the Deposition of Fragments: The Case of Micro- and Nanoplastics. *Environ. Sci. Technol.* **2024**, *58* (20), 8878–8888. <https://doi.org/10.1021/acs.est.3c07656>.

- (8) Hofmann, T.; Ghoshal, S.; Tufenkji, N.; Adamowski, J. F.; Bayen, S.; Chen, Q.; Demokritou, P.; Flury, M.; Hüffer, T.; Ivleva, N. P.; Ji, R.; Leask, R. L.; Maric, M.; Mitrano, D. M.; Sander, M.; Pahl, S.; Rillig, M. C.; Walker, T. R.; White, J. C.; Wilkinson, K. J. Plastics Can Be Used More Sustainably in Agriculture. *Commun Earth Environ* **2023**, *4* (1), 1–11. <https://doi.org/10.1038/s43247-023-00982-4>.
- (9) Koestel, J.; Schlüter, S. Quantification of the Structure Evolution in a Garden Soil over the Course of Two Years. *Geoderma* **2019**, *338*, 597–609. <https://doi.org/10.1016/j.geoderma.2018.12.030>.
- (10) Keller, T.; Colombi, T.; Ruiz, S.; Schymanski, S. J.; Weisskopf, P.; Koestel, J.; Sommer, M.; Stadelmann, V.; Breitenstein, D.; Kirchgessner, N.; Walter, A.; Or, D. Soil Structure Recovery Following Compaction: Short-Term Evolution of Soil Physical Properties in a Loamy Soil. *Soil Science Society of America Journal* **2021**, *85* (4), 1002–1020. <https://doi.org/10.1002/saj2.20240>.
- (11) Corradini, F.; Meza, P.; Eguiluz, R.; Casado, F.; Huerta-Lwanga, E.; Geissen, V. Evidence of Microplastic Accumulation in Agricultural Soils from Sewage Sludge Disposal. *Science of The Total Environment* **2019**, *671*, 411–420. <https://doi.org/10.1016/j.scitotenv.2019.03.368>.
- (12) Li, S.; Ding, F.; Flury, M.; Wang, Z.; Xu, L.; Li, S.; Jones, D. L.; Wang, J. Macro- and Microplastic Accumulation in Soil after 32 Years of Plastic Film Mulching. *Environmental Pollution* **2022**, *300*, 118945. <https://doi.org/10.1016/j.envpol.2022.118945>.
- (13) Yang, J.; Li, L.; Li, R.; Xu, L.; Shen, Y.; Li, S.; Tu, C.; Wu, L.; Christie, P.; Luo, Y. Microplastics in an Agricultural Soil Following Repeated Application of Three Types of Sewage Sludge: A Field Study. *Environmental Pollution* **2021**, *289*, 117943. <https://doi.org/10.1016/j.envpol.2021.117943>.
- (14) Li, H.-Z.; Zhu, D.; Lindhardt, J. H.; Lin, S.-M.; Ke, X.; Cui, L. Long-Term Fertilization History Alters Effects of Microplastics on Soil Properties, Microbial Communities, and Functions in Diverse Farmland Ecosystem. *Environ. Sci. Technol.* **2021**, *55* (8), 4658–4668. <https://doi.org/10.1021/acs.est.0c04849>.
- (15) Koskei, K.; Munyasya, A. N.; Wang, Y.-B.; Zhao, Z.-Y.; Zhou, R.; Indoshi, S. N.; Wang, W.; Cheruiyot, W. K.; Mburu, D. M.; Nyende, A. B.; Xiong, Y.-C. Effects of Increased Plastic Film Residues on Soil Properties and Crop Productivity in Agro-Ecosystem. *Journal of Hazardous Materials* **2021**, *414*, 125521. <https://doi.org/10.1016/j.jhazmat.2021.125521>.
- (16) Hu, Q.; Li, X.; Gonçalves, J. M.; Shi, H.; Tian, T.; Chen, N. Effects of Residual Plastic-Film Mulch on Field Corn Growth and Productivity. *Science of The Total Environment* **2020**, *729*, 138901. <https://doi.org/10.1016/j.scitotenv.2020.138901>.
- (17) Guo, J.-J.; Huang, X.-P.; Xiang, L.; Wang, Y.-Z.; Li, Y.-W.; Li, H.; Cai, Q.-Y.; Mo, C.-H.; Wong, M.-H. Source, Migration and Toxicology of Microplastics in Soil. *Environment International* **2020**, *137*, 105263. <https://doi.org/10.1016/j.envint.2019.105263>.
- (18) Gabet, E. J.; Reichman, O. J.; Seabloom, E. W. The Effects of Bioturbation on Soil Processes and Sediment Transport. *Annual Review of Earth and Planetary Sciences* **2003**, *31* (Volume 31, 2003), 249–273. <https://doi.org/10.1146/annurev.earth.31.100901.141314>.
- (19) Li, H.; Lu, X.; Wang, S.; Zheng, B.; Xu, Y. Vertical Migration of Microplastics along Soil Profile under Different Crop Root Systems. *Environmental Pollution* **2021**, *278*, 116833. <https://doi.org/10.1016/j.envpol.2021.116833>.
- (20) Tagg, A. S.; Brandes, E.; Fischer, F.; Fischer, D.; Brandt, J.; Labrenz, M. Agricultural Application of Microplastic-Rich Sewage Sludge Leads to Further Uncontrolled Contamination. *Science of The Total Environment* **2022**, *806*, 150611. <https://doi.org/10.1016/j.scitotenv.2021.150611>.
- (21) Scheurer, M.; Bigalke, M. Microplastics in Swiss Floodplain Soils. *Environ. Sci. Technol.* **2018**, *52* (6), 3591–3598. <https://doi.org/10.1021/acs.est.7b06003>.
- (22) Fuller, S.; Gautam, A. A Procedure for Measuring Microplastics Using Pressurized Fluid Extraction. *Environ. Sci. Technol.* **2016**, *50* (11), 5774–5780. <https://doi.org/10.1021/acs.est.6b00816>.
- (23) *Plastik in der Landwirtschaft: Stand des Wissens und Handlungsempfehlungen für die landwirtschaftliche Forschung, Praxis, Industrie und Behörden.* <https://ira.agroscope.ch/en-US/publication/42595> (accessed 2021-08-30).
- (24) Capowiez, Y.; Marchán, D.; Decaëns, T.; Hedde, M.; Bottinelli, N. Let Earthworms Be Functional - Definition of New Functional Groups Based on Their Bioturbation Behavior. *Soil Biology and Biochemistry* **2024**, *188*, 109209. <https://doi.org/10.1016/j.soilbio.2023.109209>.

- (25) Rillig, M. C.; Ingrassia, R.; de Souza Machado, A. A. Microplastic Incorporation into Soil in Agroecosystems. *Front. Plant Sci.* **2017**, *8*. <https://doi.org/10.3389/fpls.2017.01805>.
- (26) Frehland, S.; Kaegi, R.; Hufenus, R.; Mitrano, D. M. Long-Term Assessment of Nanoplastic Particle and Microplastic Fiber Flux through a Pilot Wastewater Treatment Plant Using Metal-Doped Plastics. *Water Research* **2020**, *182*, 115860. <https://doi.org/10.1016/j.watres.2020.115860>.
- (27) Tophinke, A. H.; Joshi, A.; Baier, U.; Hufenus, R.; Mitrano, D. M. Systematic Development of Extraction Methods for Quantitative Microplastics Analysis in Soils Using Metal-Doped Plastics. *Environmental Pollution* **2022**, *311*, 119933. <https://doi.org/10.1016/j.envpol.2022.119933>.

Review of Manuscript:

Minimal vertical transport of microplastics in soil over two years with little impact of plastics on soil macropore networks

Corresponding author: denise.mitrano@usys.ethz.ch

We thank the reviewers for their positive feedback and appreciation of our revisions. We are pleased that the clarifications and changes have addressed the comments and helped to improve the manuscript. We carefully considered the remaining point raised by Reviewer #2 and made a minor revision accordingly. Below we have answered each comment in blue, where specific additions to the manuscript text are also included into the response to the reviewers, where appropriate, in italic font. In the response to reviewers' letter, we have also included references to line numbers where these additions can be found, where line numbers refer to the new, clean version of the manuscript.

Reviewer: 2

Comment:

The author nicely addressed most of my comments, well done. There is one comment open where I kindly disagree, I suggest a minor revision:

Line 51: The point about synthetic origin is unclear and needs precision. What are these "important questions"? Why is the synthetic nature of plastics the main issue? Would biobased materials present less of a problem? Please clarify these points, as the current phrasing is vague.

Rebuttal:

The synthetic nature of plastic prevents them from biodegradation. Unlike natural materials, MPs are not biodegradable, which means they can accumulate in soil and affect ecosystems over time. Biobased materials may improve biodegradability to some extent, although not all degrade as intended. We rephrased the text (line 55-56) to convey the message more clearly and avoid confusion about synthetic origin: The resistance of MPs to biodegradation contributes to their persistence in the environment and may trigger unforeseen ecological changes.

I still do not agree with this suggestion:

Whether a plastic is biodegradable or not is solely a function of its molecular backbone and whether the bonds can be enzymatically broken (not if something is "synthetic"). Biodegradability has nothing to do with whether the source is based on renewable feedstock or fossil-based. For example, plastics based on renewable feedstock, such as sugarcane-derived polyethylene, are not biodegradable, while fossil-based PBAT, one of the major biodegradable polymers, is produced from fossil sources. Regardless, all of these plastics are produced synthetically.

This requires clarification by the authors: Plastics can be derived from fossil sources or renewable feedstocks, and they can either be biodegradable or non-biodegradable. While some authors argue that plastics based on renewable feedstock have a smaller ecological footprint, a) producing over 500 million tonnes of plastic from natural feedstocks would cause enormous harm to ecosystems, and b) the primary emissions stem from polymer production, not from the feedstock itself.

These aspects need clarification in the manuscript. The statement, "the synthetic nature of plastic prevents them from biodegradation, unlike natural materials," is, in my opinion, still incorrect.

First, I suggest you clarify the definitions of "based on renewable feedstock" and "biodegradable" in the paper. Second, I recommend including a sentence such as:

"Conventional non-biodegradable microplastics are persistent in the environment and may trigger unforeseen ecological changes."

We thank the reviewer for their overall positive feedback about our revised manuscript. We agree with the reviewer's comment that the biodegradability of plastics is determined by their molecular structure and not by their origin from renewable or fossil-based feedstocks. We have revised the manuscript to clarify that biodegradability depends on the polymer backbone and enzymatic degradability, regardless of whether the material is derived from renewable or fossil sources. We believe that explicit definitions of "renewable feedstock" and "biodegradable" are not essential for the scope of the manuscript.

We revised the manuscript and added lines 93-96:

The persistence of MPs in the environment is primarily determined by their molecular backbone and its resistance to enzymatic cleavage, which governs their biodegradability. Therefore, conventional non-biodegradable MPs are persistent in the environment and may trigger unforeseen ecological changes.